# Sequential Predictive Two-Sample and Independence Testing

**Aleksandr Podkopaev**
Walmart Global Tech
sasha.podkopaev@walmart.com

**Aaditya Ramdas**
Carnegie Mellon University
aramdas@cmu.edu

## Abstract

We study the problems of sequential nonparametric two-sample and independence testing. Sequential tests process data online and allow using observed data to decide whether to stop and reject the null hypothesis or to collect more data, while maintaining type I error control. We build upon the principle of (nonparametric) testing by betting, where a gambler places bets on future observations and their wealth measures evidence against the null hypothesis. While recently developed kernel-based betting strategies often work well on simple distributions, selecting a suitable kernel for high-dimensional or structured data, such as images, is often nontrivial. To address this drawback, we design prediction-based betting strategies that rely on the following fact: if a sequentially updated predictor starts to consistently determine (a) which distribution an instance is drawn from, or (b) whether an instance is drawn from the joint distribution or the product of the marginal distributions (the latter produced by external randomization), it provides evidence against the two-sample or independence nulls respectively. We empirically demonstrate the superiority of our tests over kernel-based approaches under structured settings. Our tests can be applied beyond the case of independent and identically distributed data, remaining valid and powerful even when the data distribution drifts over time.

## 1 Introduction

We consider two closely-related problems of nonparametric two-sample and independence testing. In the former, given observations from two distributions $P$ and $Q$, the goal is to test the null hypothesis that the distributions are the same: $H_0 : P = Q$, against the alternative that they are not: $H_1 : P \neq Q$. In the latter, given observations from some joint distribution $P_{XY}$, the goal is to test the null hypothesis that the random variables are independent: $H_0 : P_{XY} = P_X \times P_Y$, against the alternative that they are not: $H_1 : P_{XY} \neq P_X \times P_Y$. Kernel tests, such as kernel-MMD [Gretton et al., 2012] for two-sample and HSIC [Gretton et al., 2005] for independence testing, are amongst the most popular methods for solving these tasks which work well on data from simple distributions. However, their performance is sensitive to the choice of a kernel and respective parameters, like bandwidth, and applying such tests requires additional effort. Further, selecting kernels for structured data, like images, is a nontrivial task. Lastly, kernel tests suffer from decaying power in high dimensions [Ramdas et al., 2015].

Predictive two-sample and independence tests (2STs and ITs respectively) aim to address such limitations of kernelized approaches. The idea of using classifiers for two-sample testing dates back to Friedman [2004] who proposed using the output scores as a dimension reduction method. More recent works focused on the direct evaluation of a learned model for testing. In an initial arXiv 2016 preprint, Kim et al. [2021] proposed and analyzed predictive 2STs based on sample-splitting, namely testing whether the accuracy of a model trained on the first split of data and estimated on the second split is significantly better than chance. The authors established the consistency of asymptotic and exact tests in high-dimensional settings and provided rates for the case of Gaussian

37th Conference on Neural Information Processing Systems (NeurIPS 2023).

distributions. Inspired by this work, Lopez-Paz and Oquab [2017] soon after demonstrated that empirically predictive 2STs often outperform state-of-the-art 2STs, such as kernel-MMD. Recently, Hediger et al. [2022] proposed a related test that utilizes out-of-bag predictions for bagging-based classifiers, such as random forests. To incorporate measures of model confidence, many authors have also explored using test statistics that are based on the output scores instead of the binary class predictions [Kim et al., 2019, Liu et al., 2020, Cheng and Cloninger, 2022, Kübler et al., 2022].

The focus of the above works is on *batch* tests which are calibrated to have a fixed false positive rate (say, $5\%$) if the sample size is specified in advance. In contrast, we focus on the setting of sequentially released data. Our tests allow on-the-fly decision-making: an analyst can use observed data to decide whether to stop and reject the null or to collect more data, without inflating the false alarm rate.

**Problem Setup.** First, we define the problems of sequential two-sample and independence testing.

**Definition 1** (Sequential two-sample testing). Suppose that we observe a stream of i.i.d. observations $((Z_t, W_t))_{t \geq 1}$, where $W_t \sim \mathrm{Rademacher}(1/2)$, the distribution of $Z_t \mid W_t = +1$ is denoted $P$, and that of $Z_t \mid W_t = -1$ is denoted $Q$. The goal is to design a sequential test for

$$H_0 : P = Q, \tag{1a}$$
$$H_1 : P \neq Q. \tag{1b}$$

**Definition 2** (Sequential independence testing). Suppose that we observe that a stream of observations: $((X_t, Y_t))_{t \geq 1}$, where $(X_t, Y_t) \sim P_{XY}$ for $t \geq 1$. The goal is to design a sequential test for

$$H_0 : (X_t, Y_t) \sim P_{XY} \text{ and } P_{XY} = P_X \times P_Y, \tag{2a}$$
$$H_1 : (X_t, Y_t) \sim P_{XY} \text{ and } P_{XY} \neq P_X \times P_Y. \tag{2b}$$

We operate in the framework of "power-one tests" [Darling and Robbins, 1968] and define a level-$\alpha$ sequential test as a mapping $\Phi : \cup_{t=1}^{\infty} \mathcal{Z}^t \to \{0, 1\}$ that satisfies: $\mathbb{P}_{H_0}(\exists t \geq 1 : \Phi(Z_1, \dots, Z_t) = 1) \leq \alpha$. We refer to such notion of type I error control as *time-uniform*. Here, 0 stands for "do not reject the null yet" and 1 stands for "reject the null and stop". Defining the stopping time as the first time that the test outputs 1: $\tau := \inf\{t \geq 1 : \Phi(Z_1, \dots, Z_t) = 1\}$, a sequential test must satisfy

$$\mathbb{P}_{H_0}(\tau < \infty) \leq \alpha. \tag{3}$$

We aim to design *consistent* tests which are guaranteed to stop if the alternative happens to be true:

$$\mathbb{P}_{H_1}(\tau < \infty) = 1. \tag{4}$$

**Related Works.** Our construction follows the principle of testing by betting [Shafer, 2021]. The most closely related work is that of "nonparametric 2ST by betting" of Shekhar and Ramdas [2023], which later inspired several follow-up works, including sequential (marginal) kernelized independence tests of Podkopaev et al. [2023], and the sequential conditional independence tests under the model-X assumption of Grünwald et al. [2023] and Shaer et al. [2023]. We extend the line of work of Shekhar and Ramdas [2023] and of Podkopaev et al. [2023], studying predictive approaches in detail.

Sequential predictive 2STs were studied by Lhéritier and Cazals [2018, 2019], but in practice, those tests were found to be inferior to the ones developed by Shekhar and Ramdas [2023]. Recently, Pandeva et al. [2022] proposed a predictive 2ST that handles unknown class proportions using ideas from [Wasserman et al., 2020]. In Section 2, we show that our tests are closely related to [Lhéritier and Cazals, 2018, 2019, Pandeva et al., 2022], but are consistent under much milder assumptions.

**Sequential Nonparametric Two-Sample and Independence Testing by Betting.** Suppose that one observes a sequence of random variables $(Z_t)_{t \geq 1}$, where $Z_t \in \mathcal{Z}$. The principle of testing by betting [Shafer and Vovk, 2019, Shafer, 2021] can be described as follows. A player starts the game with initial capital $\mathcal{K}_0 = 1$. At round $t$, she selects a payoff function $f_t : \mathcal{Z} \to [-1, \infty)$ that satisfies $\mathbb{E}_{Z \sim P_Z}[f_t(Z) \mid \mathcal{F}_{t-1}] = 0$ for all $P_Z \in H_0$, where $\mathcal{F}_{t-1} = \sigma(Z_1, \dots, Z_{t-1})$ denotes the sigma-field generated by $Z_1, \dots, Z_{t-1}$ with $\mathcal{F}_0$ being the trivial sigma-field, and bets a fraction of her wealth $\lambda_t \mathcal{K}_{t-1}$ for an $\mathcal{F}_{t-1}$-measurable $\lambda_t \in [0, 1]$. Once $Z_t$ is revealed, her wealth is updated as

$$\mathcal{K}_t = \mathcal{K}_{t-1} + \lambda_t \mathcal{K}_{t-1} f_t(Z_t) = \mathcal{K}_{t-1}(1 + \lambda_t f_t(Z_t)). \tag{5}$$

The wealth is used to measure the evidence against $H_0$: if a player can make money in such game, the null is rejected. Formally, for testing $H_0$ at level $\alpha \in (0, 1)$, we use the stopping rule:

$$\tau = \inf\{t \geq 1 : \mathcal{K}_t \geq 1/\alpha\}. \tag{6}$$

The validity of the test follows from Ville's inequality [Ville, 1939], a time-uniform generalization of Markov's inequality, since $(\mathcal{K}_t)_{t\geq 0}$ is a nonnegative martingale starting at 1 under any $P_Z \in H_0$. To ensure high power, one has to choose $(f_t)_{t\geq 1}$ and $(\lambda_t)_{t\geq 1}$ to guarantee the growth of the wealth if the alternative is true. In the context of two-sample and independence testing, Shekhar and Ramdas [2023] and Podkopaev et al. [2023] recently proposed effective betting strategies based on kernelized measures of statistical distance and dependence respectively which admit a variational representation. In a nutshell, datapoints observed prior to a given round are used to estimate the *witness* function — one that best highlights the discrepancy between $P$ and $Q$ for two-sample (or between $P_{XY}$ and $P_X \times P_Y$ for independence) testing — and a bet is formed as an estimator of a chosen measure of distance (or dependence). In contrast, our bets are based on evaluating the performance of a sequentially learned predictor that distinguishes between instances from distributions of interest.

*Remark* 1. In practical settings, an analyst may not be able to continue collecting data forever and may adaptively stop the experiment before the wealth exceeds $1/\alpha$. In such case, one may use a different threshold for rejecting the null at a stopping time $\tau$, namely $U/\alpha$, where $U$ is a (stochastically larger than) uniform random variable on $[0,1]$ drawn independently from $(\mathcal{F}_t)_{t\geq 0}$. This choice strictly improves the power of the test without violating the validity; see [Ramdas and Manole, 2023].

**Contributions.** In Section 2, we develop sequential predictive two-sample and independence tests. We establish sufficient conditions for consistency of our tests and relate those to evaluation metrics of the underlying models. In Section 3, we conduct an extensive empirical study on synthetic and real data, justifying the superiority of our tests over the kernelized ones on structured data.

## 2 Classification-based Two-Sample Testing

We begin with the two-sample testing setting outlined in Definition 5. Let $\mathcal{G} : \mathcal{Z} \to [-1, 1]$ denote a class of predictors used to distinguish between instances from $P$ (labeled as $+1$) and $Q$ (labeled as $-1$)[1]. We assume that: (a) if $g \in \mathcal{G}$, then $-g \in \mathcal{G}$, (b) if $g \in \mathcal{G}$ and $s \in [0, 1]$, then $sg \in \mathcal{G}$, and (c) predictions are based on $\text{sign}[g(\cdot)]$, and if $g(z) = 0$, then $z$ is assigned to the positive class. Two natural evaluation metrics of a predictor $g \in \mathcal{G}$ include the misclassification and the squared risks:

$$R_{\mathrm{m}}(g) := \mathbb{P}\left(W \cdot \text{sign}\left[g\left(Z\right)\right] < 0\right), \quad R_{\mathrm{s}}(g) := \mathbb{E}\left[\left(g(Z) - W\right)^2\right], \tag{7}$$

which give rise to the following measures of distance between $P$ and $Q$, namely

$$d_{\mathrm{m}}(P,Q) := \sup_{g\in\mathcal{G}}\left(\tfrac{1}{2} - R_{\mathrm{m}}(g)\right), \quad d_{\mathrm{s}}(P,Q) := \sup_{g\in\mathcal{G}}\left(1 - R_{\mathrm{s}}(g)\right). \tag{8}$$

It is easy to check that $d_{\mathrm{m}}(P,Q) \in [0, 1/2]$ and $d_{\mathrm{m}}(P,Q) \in [0, 1]$. The upper bounds hold due to the non-negativity of the risks and the lower bounds follow by considering $g : g(z) = 0, \forall z \in \mathcal{Z}$. Note that the misclassification risk is invariant to rescaling ($R_{\mathrm{m}}(sg) = R_{\mathrm{m}}(g), \forall s \in (0, 1]$), whereas the squared risk is not, and rescaling any $g$ to optimize the squared risk provides better contrast between $P$ and $Q$. In the next result, whose proof is deferred to Appendix D.3, we present an important relationship between the squared risk of a rescaled predictor and its expected margin: $\mathbb{E}\left[W \cdot g(Z)\right]$.

**Proposition 1.** *Fix an arbitrary predictor $g \in \mathcal{G}$. The following claims hold[2]:*

*1. For the misclassification risk, we have that:*

$$\sup_{s\in[0,1]}\left(\tfrac{1}{2} - R_{\mathrm{m}}(sg)\right) = \left(\tfrac{1}{2} - R_{\mathrm{m}}(g)\right) \vee 0 = \left(\tfrac{1}{2} \cdot \mathbb{E}\left[W \cdot \text{sign}\left[g(Z)\right]\right]\right) \vee 0. \tag{9}$$

*2. For the squared risk, we have that:*

$$\sup_{s\in[0,1]}\left(1 - R_{\mathrm{s}}(sg)\right) \geq \left(\mathbb{E}\left[W \cdot g(Z)\right] \vee 0\right) \cdot \left(\frac{\mathbb{E}\left[W \cdot g(Z)\right]}{\mathbb{E}\left[g^2(Z)\right]} \wedge 1\right) \tag{10}$$

*Further, $d_{\mathrm{s}}(P,Q) > 0$ if and only if there exists $g \in \mathcal{G}$ such that $\mathbb{E}\left[W \cdot g(Z)\right] > 0$.*

---

[1] Similar argument can be applied to general scoring-based classifiers: $g : \mathcal{Z} \to \mathbb{R}$, e.g., SVMs, by considering $\tilde{\mathcal{G}} = \{\tilde{g} : \tilde{g}(z) = \tanh(s \cdot g(z)), \ g \in \mathcal{G}, \ s > 0\}$, where the constant $s > 0$ corrects the scale of the scores.

[2] $\vee$ and $\wedge$ denote a maximum and a minimum respectively: $a \vee b = \max\{a, b\}$, $a \wedge b = \min\{a, b\}$.

Consider an arbitrary predictor $g \in \mathcal{G}$. Note that under the null $H_0$ in (1a), the misclassification risk $R_{\mathrm{m}}(g)$ does not depend on $g$, being equal to 1/2, whereas the squared risk $R_{\mathrm{s}}(g)$ does. In contrast, the lower bound (10) no longer depends on $g$ under the null $H_0$, being equal to 0.

**Oracle Test.** It is a known fact that the minimizer of either the misclassification or the squared risk is $g^{\mathrm{Bayes}}(z) = 2\eta(z) - 1$, where $\eta(z) = \mathbb{P}(W = +1 \mid Z = z)$. Since $g^{\mathrm{Bayes}}$ may not belong to $\mathcal{G}$, we consider $g_\star \in \mathcal{G}$, which minimizes either the misclassification or the squared risk over predictors in $\mathcal{G}$, and omit superscripts for brevity. To design payoff functions, we follow Proposition 1 and consider

$$f_\star^{\mathrm{m}}(Z_t, W_t) = W_t \cdot \mathrm{sign}\left[g_\star(Z_t)\right] \in \{-1, 1\}, \tag{11a}$$

$$f_\star^{\mathrm{s}}(Z_t, W_t) = W_t \cdot g_\star(Z_t) \in [-1, 1]. \tag{11b}$$

Let the *oracle* wealth processes based on misclassification and squared risks $(\mathcal{K}_t^{\mathrm{m},\star})_{t \geq 0}$ and $(\mathcal{K}_t^{\mathrm{s},\star})_{t \geq 0}$ be defined by using the payoff functions (11a) and (11b) respectively, along with a predictable sequence of betting fractions $(\lambda_t)_{t \geq 1}$ selected via online Newton step (ONS) strategy [Hazan et al., 2007] (Algorithm 1), which has been studied in the context of coin-betting by Cutkosky and Orabona [2018]. If a constant betting fraction is used throughout: $\lambda_t = \lambda$, $\forall t$, then

$$\mathbb{E}\left[\tfrac{1}{t} \log \mathcal{K}_t^{i,\star}\right] = \mathbb{E}\left[\log(1 + \lambda f_\star^i(Z, W))\right], \quad i \in \{\mathrm{m}, \mathrm{s}\}. \tag{12}$$

To illustrate the tightness of our results, we consider the optimal constant betting fractions which maximize the log-wealth (12) and are constrained to lie in $[-0.5, 0.5]$, like ONS bets:

$$\lambda_\star^i = \underset{\lambda \in [-0.5, 0.5]}{\arg\max} \ \mathbb{E}\left[\log(1 + \lambda f_\star^i(Z, W))\right], \quad i \in \{\mathrm{m}, \mathrm{s}\}. \tag{13}$$

---

**Algorithm 1** Online Newton step (ONS) strategy for selecting betting fractions

---

**Input:** sequence of payoffs $(f_t)_{t \geq 1}$, $\lambda_1^{\mathrm{ONS}} = 0$, $a_0 = 1$.
**for** $t = 1, 2, \ldots$ **do**
    Observe $f_t \in [-1, 1]$;
    Set $z_t := f_t / (1 + \lambda_t^{\mathrm{ONS}} f_t)$;
    Set $a_t := a_{t-1} + z_t^2$;
    Set $\lambda_{t+1}^{\mathrm{ONS}} := \frac{1}{2} \wedge \left(0 \vee \left(\lambda_t^{\mathrm{ONS}} + \frac{2}{2 - \log 3} \cdot \frac{z_t}{a_t}\right)\right)$;

---

Assuming that the 2ST null is false and a predictor with non-trivial prediction risk is used, it is easy to see that the optimal constant betting fraction has to be positive, which is why in Algorithm 1 we truncate $\lambda_t^{\mathrm{ONS}}$ at 0 instead of $-1/2$ (the latter option is used in [Cutkosky and Orabona, 2018]). We note that this is allowed and does not affect any theoretical claims of the paper. In early simulations, we also did not observe any major differences between the two truncating options. We conclude with the following result for the oracle tests, whose proof is deferred to Appendix D.3.

**Theorem 1.** *The following claims hold:*

1. *Suppose that $H_0$ in (1a) is true. Then the oracle sequential test based on either $(\mathcal{K}_t^{\mathrm{m},\star})_{t \geq 0}$ or $(\mathcal{K}_t^{\mathrm{s},\star})_{t \geq 0}$ ever stops with probability at most $\alpha$: $\mathbb{P}_{H_0}(\tau < \infty) \leq \alpha$.*

2. *Suppose that $H_1$ in (1b) is true. Then:*

   (a) *The growth rate of the oracle wealth process $(\mathcal{K}_t^{\mathrm{m},\star})_{t \geq 0}$ satisfies:*

   $$\liminf_{t \to \infty} \left(\tfrac{1}{t} \log \mathcal{K}_t^{\mathrm{m},\star}\right) \overset{\mathrm{a.s.}}{\geq} \left(\tfrac{1}{2} - R_{\mathrm{m}}(g_\star)\right)^2. \tag{14}$$

   *If $R_{\mathrm{m}}(g_\star) < 1/2$, then the test based on $(\mathcal{K}_t^{\mathrm{m},\star})_{t \geq 0}$ is consistent: $\mathbb{P}_{H_1}(\tau < \infty) = 1$. Further, the optimal growth rate achieved by $\lambda_\star^{\mathrm{m}}$ in (13) satisfies:*

   $$\mathbb{E}\left[\log(1 + \lambda_\star^{\mathrm{m}} f_\star^{\mathrm{m}}(Z, W))\right] \leq \left(\tfrac{16}{3} \cdot \left(\tfrac{1}{2} - R_{\mathrm{m}}(g_\star)\right)^2 \wedge \left(\tfrac{1}{2} - R_{\mathrm{m}}(g_\star)\right)\right). \tag{15}$$

   (b) *The growth rate of the oracle wealth process $(\mathcal{K}_t^{\mathrm{s},\star})_{t \geq 0}$ satisfies:*

   $$\liminf_{t \to \infty} \left(\tfrac{1}{t} \log \mathcal{K}_t^{\mathrm{s},\star}\right) \overset{\mathrm{a.s.}}{\geq} \tfrac{1}{4} \cdot \mathbb{E}\left[W \cdot g_\star(Z)\right]. \tag{16}$$

*If $\mathbb{E}\left[W \cdot g_\star(Z)\right] > 0$, then the test based on $(\mathcal{K}_t^{\mathrm{s},\star})_{t\geq 0}$ is consistent: $\mathbb{P}_{H_1}\left(\tau < \infty\right) = 1$. Further, the optimal growth rate achieved by $\lambda_\star^{\mathrm{s}}$ in (13) satisfies:*

$$\mathbb{E}\left[\log(1 + \lambda_\star^{\mathrm{s}} f_\star^{\mathrm{s}}(Z, W))\right] \leq \tfrac{1}{2} \cdot \mathbb{E}\left[W \cdot g_\star(Z)\right]. \tag{17}$$

Theorem 1 precisely characterizes the properties of the oracle wealth processes and relates those to interpretable metrics of predictive performance. Further, the proof of Theorem 1 highlights a direct impact of the variance of the payoffs on the wealth growth rate, and hence the power of the resulting sequential tests (as the null is rejected once the wealth exceeds $1/\alpha$).

The second moment of the payoffs based on the misclassification risk (11a) is equal to one, resulting in a *slow* growth: the bound (14) is proportional to *squared* deviation of the misclassification risk from one half. The bound (15) shows that the growth rate with the ONS strategy matches, up to constants, that of the oracle betting fraction. Note that the second term in (15) characterizes the growth rate if $R_{\mathrm{m}}(g_\star) < 5/16$ (low Bayes risk). In this regime, the growth rate of our test is at least $(3/16) \cdot (1/2 - R_{\mathrm{m}}(g_\star))$ which is close to the optimal rate. The second moment of the payoffs based on the squared risk is more insightful. First, we present a result for the case when the oracle predictor $g_\star$ in (11b) is replaced by an arbitrary $g \in \mathcal{G}$. The proof is deferred to Appendix D.3.

**Corollary 1.** *Consider an arbitrary $g \in \mathcal{G}$ with nonnegative expected margin: $\mathbb{E}\left[W \cdot g(Z)\right] \geq 0$. Then the growth rate of the corresponding wealth process $(\mathcal{K}_t^{\mathrm{s}})_{t\geq 0}$ satisfies:*

$$\liminf_{t\to\infty} \left(\tfrac{1}{t} \log \mathcal{K}_t^{\mathrm{s}}\right) \overset{\mathrm{a.s.}}{\geq} \tfrac{1}{4}\left(\sup_{s\in[0,1]} \left(1 - R_{\mathrm{s}}(sg)\right) \wedge \mathbb{E}\left[W \cdot g(Z)\right]\right) \tag{18a}$$

$$\geq \tfrac{1}{4}\left(\mathbb{E}\left[W \cdot g(Z)\right]\right)^2, \tag{18b}$$

*and the optimal growth rate achieved by $\lambda_\star^{\mathrm{s}}$ in (13) satisfies:*

$$\mathbb{E}\left[\log(1 + \lambda_\star^{\mathrm{s}} f^{\mathrm{s}}(Z, W))\right] \leq \left(\tfrac{4}{3} \cdot \sup_{s\in[0,1]} \left(1 - R_{\mathrm{s}}\left(sg\right)\right)\right) \wedge \left(\tfrac{1}{2} \cdot \mathbb{E}\left[W \cdot g(Z)\right]\right). \tag{19}$$

Corollary 1 states that for an arbitrary $g \in \mathcal{G}$, the growth rate is lower bounded by the minimum of the expected margin and the (optimized) squared risk of such predictor. While the latter term is always smaller for the optimal $g_\star$, this may not hold for an arbitrary $g \in \mathcal{G}$. The lower bound (18b), which follows from Proposition 1, is always worse than that for $g_\star$ (the expected margin is squared). The upper bound (19) shows that the growth rate with the ONS strategy matches, up to constants, that of the optimal constant betting fraction. Before presenting a practical sequential 2ST, we provide two important remarks that further contextualize the current work in the literature.

*Remark* 2. In practice, we learn a predictor sequentially and have to choose a learning algorithm. Note that (18a) suggests that direct margin maximization may hurt the power of the resulting 2ST: the squared risk is sensitive to miscalibrated and overconfident predictors. Kübler et al. [2022] made a similar conjecture in the context of batch two-sample testing. To optimize the power, the authors suggested minimizing the cross-entropy or the squared loss and related such approach to maximizing the signal-to-noise ratio, a heuristic approach that was proposed earlier by Sutherland et al. [2017][3].

*Remark* 3. Suppose that $g^{\mathrm{Bayes}} \in \mathcal{G}$ and consider the payoff function based on the squared risk (11b). At round $t$, the wealth of a player $\mathcal{K}_{t-1}$ is multiplied by

$$1 + \lambda_t \cdot W_t \cdot g^{\mathrm{Bayes}}(Z_t) = (1 - \lambda_t) \cdot 1 + \lambda_t \cdot \left(1 + W_t \cdot g^{\mathrm{Bayes}}(Z_t)\right)$$

$$= (1 - \lambda_t) \cdot 1 + \lambda_t \cdot \frac{(\eta(Z_t))^{\mathbb{1}\{W_t=1\}}(1 - \eta(Z_t))^{\mathbb{1}\{W_t=-1\}}}{\left(\tfrac{1}{2}\right)^{\mathbb{1}\{W_t=1\}}\left(\tfrac{1}{2}\right)^{\mathbb{1}\{W_t=-1\}}}, \tag{20}$$

and hence, the betting fractions interpolate between the regimes of not betting and betting using a likelihood ratio. From this standpoint, 2STs of Lhéritier and Cazals [2018, 2019], Pandeva et al. [2022] set $\lambda_t = 1, \forall t$, and use only the second term for updating the wealth despite the fact that the true likelihood ratio is unknown. An argument about the consistency of such test hence requires imposing strong assumptions about a sequence of predictors $(g_t)_{t\geq 1}$ [Lhéritier and Cazals, 2018, 2019]. Our test differs in a critical way: we use a sequence of betting fractions, $(\lambda_t)_{t\geq 1}$, which adapts to the quality of the underlying predictors, yielding a consistent test under much weaker assumptions.

---

[3]Standard CLT does not apply directly when the conditioning set grows; see [Kim and Ramdas, 2020].

**Example 1.** Consider $P = \mathcal{N}(0, 1)$ and $Q = \mathcal{N}(\delta, 1)$ for 20 values of $\delta$, equally spaced in $[0, 0.5]$. For a given $\delta$, the Bayes-optimal predictor is

$$g^{\text{Bayes}}(z) = \frac{\varphi(z; 0, 1) - \varphi(z; \delta, 1)}{\varphi(z; 0, 1) + \varphi(z; \delta, 1)} \in [-1, 1], \tag{21}$$

where $\varphi(z; \mu, \sigma^2)$ denotes the density of $\mathcal{N}(\mu, \sigma^2)$ evaluated at $z$. In Figure 1a, we compare tests that use (a) the Bayes-optimal predictor, (b) a predictor constructed with the plug-in estimates of the means and variances. While in the former case betting using a likelihood ratio ($\lambda_t = 1, \forall t$) is indeed optimal, our test with an adaptive sequence $(\lambda_t)_{t \geq 1}$ is superior when a predictor is learned. The difference becomes even more drastic in Figure 1b where a (regularized) $k$-NN predictor is used.

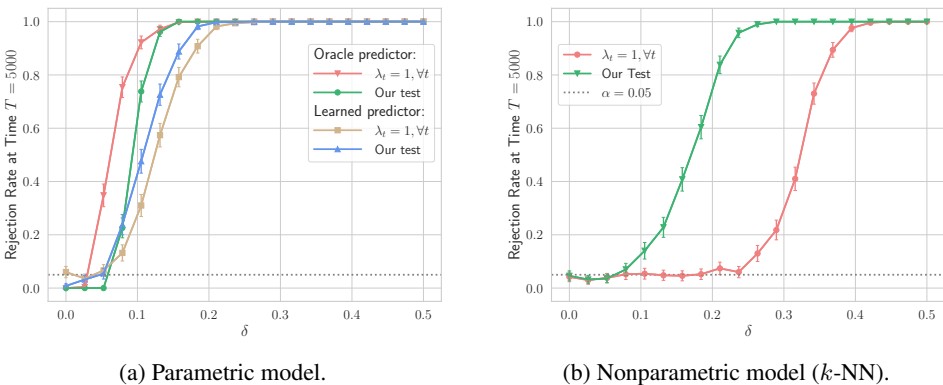

(a) Parametric model.   (b) Nonparametric model ($k$-NN).

Figure 1: Comparison between our 2ST with adaptive betting fractions and the likelihood ratio test for Example 1. While the likelihood ratio test is better if the Bayes-optimal predictor is used, our test is superior if a predictor is learned. The results are aggregated over 500 runs for each value of $\delta$.

**Sequential Classification-based 2ST (Seq-C-2ST).** Let $\mathcal{A}_c : (\cup_{t \geq 1} (\mathcal{Z} \times \{-1, +1\})^t) \times \mathcal{G} \to \mathcal{G}$ denote a learning algorithm which maps a training dataset of any size and previously used classifier, to an updated predictor. For example, $\mathcal{A}_c$ may apply a single gradient descent step using the most recent observation to update a model. We start with $\mathcal{D}_0 = \emptyset$ and $g_1 \in \mathcal{G} : g_1(z) = 0$, for any $z \in \mathcal{Z}$. At round $t$, we use one of the payoffs:

$$f_t^{\text{m}}(Z_t, W_t) = W_t \cdot \text{sign}\left[g_t(Z_t)\right] \in \{-1, 1\}, \tag{22a}$$
$$f_t^{\text{s}}(Z_t, W_t) = W_t \cdot g_t(Z_t) \in [-1, 1]. \tag{22b}$$

After $(Z_t, W_t)$ is used for betting, we update a training dataset: $\mathcal{D}_t = \mathcal{D}_{t-1} \cup \{(Z_t, W_t)\}$, and an existing predictor: $g_{t+1} = \mathcal{A}_c(\mathcal{D}_t, g_t)$. We summarize our sequential classification-based 2ST (Seq-C-2ST) in Algorithm 2. We note that using pre-trained models (e.g., for image data) as initialization may definitely improve the performance during the early stages of testing in practice.

---

**Algorithm 2** Sequential classification-based 2ST (Seq-C-2ST)

---

**Input:** level $\alpha \in (0, 1)$, data stream $((Z_t, W_t))_{t \geq 1}$, $g_1(z) \equiv 0$, $\mathcal{A}_c$, $\mathcal{D}_0 = \emptyset$, $\lambda_1^{\text{ONS}} = 0$.
**for** $t = 1, 2, \ldots$ **do**
    Evaluate the payoff $f_t^{\text{s}}(Z_t, W_t)$ as in (22a);
    Using $\lambda_t^{\text{ONS}}$, update the wealth process $\mathcal{K}_t^{\text{s}}$ as per (5);
    **if** $\mathcal{K}_t^{\text{s}} \geq 1/\alpha$ **then**
        Reject $H_0$ and stop;
    **else**
        Update the training dataset: $\mathcal{D}_t := \mathcal{D}_{t-1} \cup \{(Z_t, W_t)\}$;
        Update predictor: $g_{t+1} = \mathcal{A}_c(\mathcal{D}_t, g_t)$;
        Compute $\lambda_{t+1}^{\text{ONS}}$ (Algorithm 1) using $f_t^{\text{s}}(Z_t, W_t)$;

---

While we do not need any assumptions to confirm the type I error control, we place some mild assumptions on the learning algorithm $\mathcal{A}_c$ to argue about the consistency.

**Assumption 1** ($R_{\mathrm{m}}$-learnability). Suppose that $H_1$ in (1b) is true. An algorithm $\mathcal{A}_{\mathrm{c}}$ is such that the resulting sequence $(g_t)_{t \geq 1}$ satisfies: $\limsup_{t \to \infty} \frac{1}{t} \sum_{i=1}^{t} \mathbb{1}\{W_i \cdot \mathrm{sign}[g_i(Z_i)] < 0\} \overset{\text{a.s.}}{<} 1/2$.

**Assumption 2** ($R_{\mathrm{s}}$-learnability). Suppose that $H_1$ in (1b) is true. An algorithm $\mathcal{A}_{\mathrm{c}}$ is such that the resulting sequence $(g_t)_{t \geq 1}$ satisfies: $\limsup_{t \to \infty} \frac{1}{t} \sum_{i=1}^{t} (g_i(Z_i) - W_i)^2 \overset{\text{a.s}}{<} 1$.

In words, the above assumptions state that a sequence of predictors $(g_t)_{t \geq 1}$ is better than a chance predictor on average. We conclude with the following result, whose proof is deferred to Appendix D.3.

**Theorem 2.** *The following claims hold for Seq-C-2ST (Algorithm 2):*

1. *If $H_0$ in (1a) is true, the test ever stops with probability at most $\alpha$: $\mathbb{P}_{H_0}(\tau < \infty) \leq \alpha$.*

2. *Suppose that $H_1$ in (1b) is true. Then:*

    (a) *Under Assumption 1, the test with the payoff (22a) is consistent: $\mathbb{P}_{H_1}(\tau < \infty) = 1$.*
    (b) *Under Assumption 2, the test with the payoff (22b) is consistent: $\mathbb{P}_{H_1}(\tau < \infty) = 1$.*

*Remark* 4. Suppose that we perform two-sample testing for $d$-dimensional data. At time $T$, the total accumulated computation for the kernelized test of Shekhar and Ramdas [2023] is $O(dT^2)$. For our test, the answer depends on the chosen classifier and learning algorithm. Using logistic regression in combination with gradient descent for updating model parameters results in cheap updates and payoff evaluation (both are $O(d)$ at each round, and hence the total accumulated computation at time $T$ is $O(dT)$). For $k$-NN classifier, no parameters have to be updated, yet evaluating payoffs becomes more expensive with a growing sample size, resulting in the total accumulated computation of $O(kdT^2)$ at time $T$. For more complex models like neural nets, runtime depends on the chosen architecture: the total accumulated computation at time $T$ is $O((cB + F)T)$, where $F$ and $B$ are the costs of forward-propagation and back-propagation steps respectively and $c$ is the number of back-propagation steps applied after processing the next point (the exact cost depends on the architecture).

**Sequential Classification-based Independence Test (Seq-C-IT).** Under the setting of Definition 2, a single point from $P_{XY}$ is revealed at each round. Following [Podkopaev et al., 2023], we bet on two points from $P_{XY}$ (labeled as $+1$) and utilize external randomization to produce instances from $P_X \times P_Y$ (labeled as $-1$). Let $\mathcal{A}_{\mathrm{c}}^{\mathrm{IT}} : (\cup_{t \geq 1}((\mathcal{X} \times \mathcal{Y}) \times \{-1, +1\})^t) \times \mathcal{G} \to \mathcal{G}$ denote a learning algorithm which maps a training dataset of any size and previously used classifier, to an updated predictor. We start with $\mathcal{D}_0 = \emptyset$ and $g_1 : g_1(x, y) = 0, \forall (x, y) \in \mathcal{X} \times \mathcal{Y}$. We use derandomized versions of the payoffs (22), e.g., instead of (22b), we use

$$
\begin{aligned}
f_t^{\mathrm{s}}((X_{2t-1}, Y_{2t-1}), (X_{2t}, Y_{2t})) = {} & \tfrac{1}{4}(g_t(X_{2t-1}, Y_{2t-1}) + g_t(X_{2t}, Y_{2t})) \\
& - \tfrac{1}{4}(g_t(X_{2t-1}, Y_{2t}) + g_t(X_{2t}, Y_{2t-1})).
\end{aligned}
\tag{23}
$$

After $(X_{2t-1}, Y_{2t-1}), (X_{2t}, Y_{2t})$ have been used for betting, we update a training dataset:

$$
\mathcal{D}_t = \mathcal{D}_{t-1} \cup \{((X_{2t-1}, Y_{2t-1}), +1), ((X_{2t}, Y_{2t}), +1), ((X_{2t}, Y_{2t}), -1), ((X_{2t}, Y_{2t-1}), -1)\},
$$

and an existing predictor: $g_{t+1} = \mathcal{A}_{\mathrm{c}}^{\mathrm{IT}}(\mathcal{D}_t, g_t)$. Seq-C-IT inherits the time-uniform type I error control and the consistency guarantees of Theorem 2, and we omit details for brevity.

## 3 Experiments

**Synthetic Experiments.** In our evaluation, we first consider synthetic datasets where the complexity of the independence testing setup is characterized by a single univariate parameter. We set the monitoring horizon to $T = 5000$ points from $P_{XY}$, and for each parameter value, we aggregate the results over 200 runs. In particular, we use the following synthetic settings:

1. *Spherical model.* Let $(U_t)_{t \geq 1}$ be a sequence of random vectors on a unit sphere in $\mathbb{R}^d$: $U_t \overset{\text{iid}}{\sim} \mathrm{Unif}(\mathbb{S}^d)$, and let $u_{(i)}$ denote the $i$-th coordinate of $u$. For $t \geq 1$, we take

$$
(X_t, Y_t) = ((U_t)_{(1)}, (U_t)_{(2)}).
$$

We consider $d \in \{3, \ldots, 10\}$, where larger $d$ defines a harder setup.

2. *Hard-to-detect-dependence (HTDD) model.* We sample $((X_t, Y_t))_{t \geq 1}$ from

$$p(x,y) = \frac{1}{4\pi^2} \left(1 + \sin(wx)\sin(wy)\right) \cdot \mathbb{1}\left\{(x,y) \in [-\pi, \pi]^2\right\}. \quad (24)$$

We consider $w \in \{0, \ldots, 6\}$, where $H_0$ is true (random variables are independent) if and only if $w = 0$. For $w > 0$, $\text{Corr}(X, Y) \approx 1/w^2$, and the setup is harder for larger $w$.

For the comparison, we use two predictive models to construct Seq-C-ITs:

1. Let $\mathcal{N}_t(z) := \mathcal{N}(z, \mathcal{D}_{t-1}, k_t)$ define the set of $k_t$ closest points in $\mathcal{D}_{t-1}$ to a query point $z := (x, y)$. We consider a *regularized $k$-NN predictor:* $\hat{g}_t(z) = \frac{1}{k_t + 1} \sum_{(Z,W) \in \mathcal{N}_t(z)} W$. We select the number of neighbors using the square-root rule: $k_t = \sqrt{|\mathcal{D}_{t-1}|} = \sqrt{4(t-1)}$.
2. We use a multilayer perceptron (MLP) with three hidden layers and 128, 64 and 32 neurons respectively and the parameters learned using an incremental training scheme.

We use the HSIC-based sequential kernelized independence test (SKIT) [Podkopaev et al., 2023] as a reference test and defer details, such as MLP training scheme and SKIT hyperparameters, to Appendix E.1. In Figure 2, we observe that SKIT outperforms Seq-C-ITs under the spherical model (with no localized dependence structure), whereas, under the structured HTDD model, Seq-C-ITs, is superior. Further, inspecting Figure 2b at $w = 0$ confirms that all tests control the type I error. We refer the reader to Appendix E.2 for additional experiments on synthetic data with localized dependence where Seq-C-ITs are superior. In Appendix E.2, we also provide the results for the average *stopping times* of our tests: we empirically confirm that our tests are adaptive to the complexity of a problem at hand: they stop earlier on easy tasks and later on harder ones.

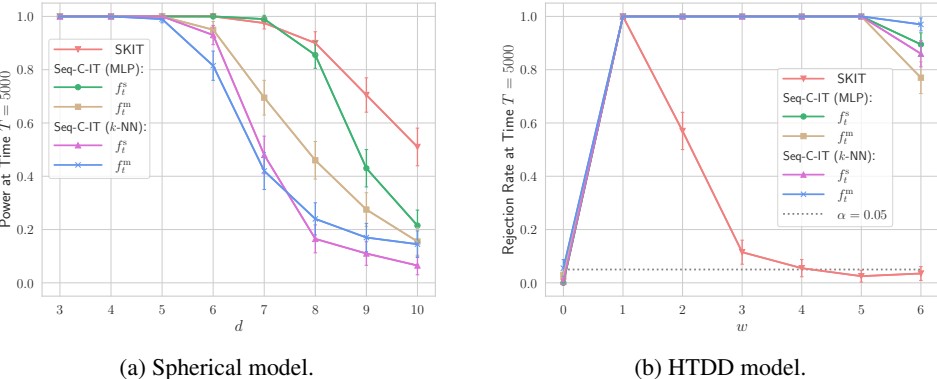

(a) Spherical model.  (b) HTDD model.

Figure 2: Power of different sequential independence tests on synthetic data from Section 3. Under the spherical model (no localized dependence), SKIT is better than Seq-C-ITs. Under the (structured) HTDD model, SKIT is inferior to sequential predictive independence tests.

**Real Data Experiments.** First, we compare sequential classification-based and kernelized 2STs using Karolinska Directed Emotional Faces dataset (KDEF) [Lundqvist et al., 1998] which contains images of actors and actresses expressing different emotions: afraid (AF), angry (AN), disgusted (DI), happy (HA), neutral (HE), sad (SA), and surprised (SU). Following earlier works [Lopez-Paz and Oquab, 2017, Jitkrittum et al., 2016], we focus on straight profile only and assign HA, NE, SU emotions to the positive class (instances from $P$), and AF, AN, DI emotions to the negative class (instances from $Q$); see Figure 3a. We remove corrupted images and obtain a dataset containing 802 images with six different emotions. The original images ($562 \times 762$ pixels) are cropped to exclude the background, resized to $64 \times 64$ pixels and converted to grayscale.

For Seq-C-2ST, we use a small CNN as an underlying model and defer details about the architecture and training to Appendix E.1. As a reference kernel-based 2ST, we use the sequential MMD test of Shekhar and Ramdas [2023] and adapt it to the setting where at each round either an observation from $P$ or that from $Q$ is revealed; see Appendix E.1 for details. We omit the comparison to Lhéritier and Cazals [2019] since their test has been shown to be inferior to the one developed by Shekhar and Ramdas [2023]. In Figure 3b, we illustrate that while both tests achieve perfect power after processing sufficiently many observations, our Seq-C-2ST requires fewer observations to do so.

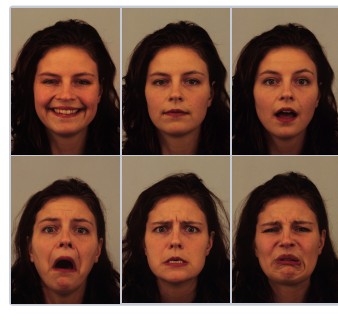
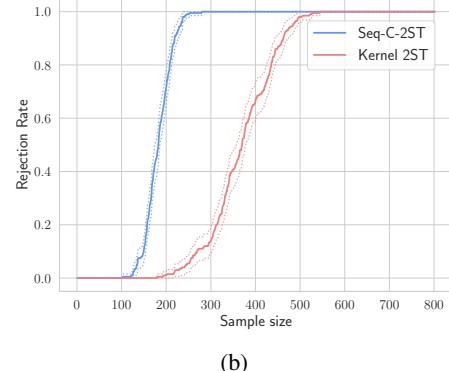

(a)                                             (b)

Figure 3: (a) Examples of instances from $P$ (top row) and $Q$ (bottom row) for KDEF dataset. (b) Rejection rates for our test (Seq-C-2ST) and the sequential kernelized 2ST. While both tests achieve perfect power with enough data, our test is superior to the kernelized approach, requiring fewer observations to do so. The results are averaged over 200 random orderings of the data.

Next, we compare two independence tests using MNIST image dataset [LeCun et al., 1998]. To simulate the null setting, we sample pairs of random images from the entire dataset, and to simulate the alternative, we sample pairs of random images depicting the same digit (Figure 4a). For Seq-C-IT, we use MLP with the same architecture as for simulations on synthetic data. For SKIT, we use the median heuristic with 20 points from $P_{XY}$ to compute kernel hyperparameters. In Figure 4b, we show that while both tests control the type I error under $H_0$, SKIT is inferior to Seq-C-IT under $H_1$, requiring twice as much data to achieve perfect power.

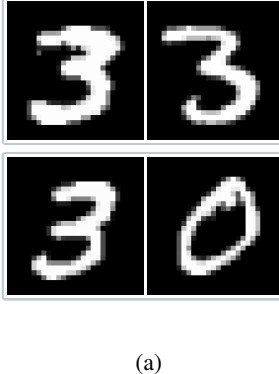
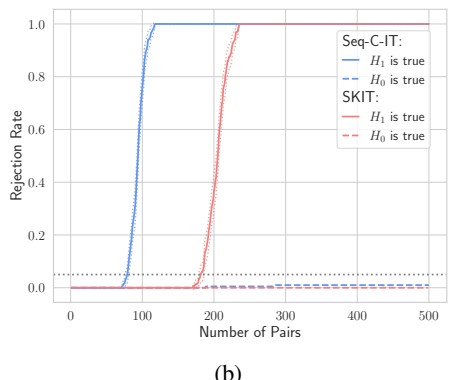

(a)                                             (b)

Figure 4: (a) Instances from the $P_{XY}$ (top row) and $P_X \times P_Y$ (bottom row) for MNIST dataset. (b) While both independence tests control the type I error under $H_0$, Seq-C-IT outperforms SKIT under $H_1$, rejecting the null much sooner. The results are aggregated over 200 runs.

## 4   Conclusion

While kernel methods are state-of-the-art for nonparametric two-sample and independence testing, their performance often deteriorates on complex data, e.g., high-dimensional data with localized dependence. In such settings, prediction-based tests are often much more effective. In this work, we developed sequential predictive two-sample and independence tests following the principle of testing by betting. Our tests control the type I error despite continuously monitoring the data and are consistent under weak and tractable assumptions. Further, our tests provably adapt to the complexity of a problem at hand: they stop earlier on easy tasks and later on harder ones. An additional advantage of our tests is that an analyst may modify the design choices, e.g., model architecture, on-the-fly. Through experiments on synthetic and real data, we confirm that our tests are competitive to kernel-based ones overall and outperform those under structured settings.

We refer the reader to the Appendix for additional results that were not included in the main paper:

1. In Appendix A, we complement classification-based ITs with a regression-based approach. Regression-based ITs represent an alternative to the classification-based approach in settings where a data stream $((X_t, Y_t))_{t \geq 1}$ may be processed directly as feature-response pairs.

2. In Section 2, we considered the case of balanced classes, meaning that at each round, an instance from either $P$ or $Q$ is observed with equal chance. In Appendix B, we extend the methodology to a more general case of two-sample testing with unknown class proportions.

3. Batch two-sample and independence tests rely on either a cutoff computed using the asymptotic null distribution of a chosen test statistic (when it is tractable) or a permutation p-value, and if the distribution drifts, both approaches fail to provide the type I error control. In contrast, Seq-C-2ST and Seq-C-IT remain valid beyond the i.i.d. setting by construction (analogous to tests developed by Shekhar and Ramdas [2023], Podkopaev et al. [2023]), and we refer the reader to Appendix C for more details.

**Acknowledgements.**    The authors thank Ian Waudby-Smith and Tudor Manole for fruitful discussions. AR acknowledges support from NSF grants IIS-2229881 and DMS-2310718.

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

# Appendix

## A  Regression-based Independence Testing

Regression-based independence tests represent an alternative to classification-based approaches in settings where a data stream $((X_t, Y_t))_{t \geq 1}$ may be processed directly as feature-response pairs. Suppose that one selects a functional class $\mathcal{G} : \mathcal{X} \to \mathcal{Y}$ for performing such prediction task, and let $\ell$ denote a loss function that evaluates the quality of predictions. For example, if $(Y_t)_{t \geq 1}$ is a sequence of univariate random variables, one can use the squared loss: $\ell(g(x), y) = (g(x) - y)^2$, or the absolute loss: $\ell(g(x), y) = |g(x) - y|$.

Such tests rely on the following idea: if the alternative $H_1$ in (2b) is true and a sequence of sequentially updated predictors $(g_t)_{t \geq 1}$ has nontrivial predictive power, then the losses on random instances drawn from the joint distribution $P_{XY}$ are expected to be less on average than the losses on random instances from $P_X \times P_Y$. For the $t$-th pair of points from $P_{XY}$, we can label the losses of $g_t$ on all possible $(X, Y)$-pairs as

$$
\begin{aligned}
L_{2t-1} &= \ell\left(g_t(X_{2t-1}), Y_{2t-1}\right), \quad L_{2t} = \ell\left(g_t(X_{2t}), Y_{2t}\right), \\
L'_{2t-1} &= \ell\left(g_t(X_{2t-1}), Y_{2t}\right), \qquad L'_{2t} = \ell\left(g_t(X_{2t}), Y_{2t-1}\right).
\end{aligned}
\tag{25}
$$

One can view this problem as sequential two-sample testing under distribution drift (due to incremental learning of $(g_t)_{t \geq 1}$). Hence, one may use either Seq-C-2ST from Section 2 or sequential kernelized 2ST of Shekhar and Ramdas [2023] on the resulting sequence of the losses on observations from $P_{XY}$ and $P_X \times P_Y$. In what follows, we analyze a direct approach where testing is performed by comparing the losses on instances drawn from the two distributions. A critical difference with a construction of Seq-C-2ST is that to design a valid betting strategy one has to ensure that the payoff functions are lower bounded by negative one.

### A.1  Proxy Regression-based Independence Test

To avoid cases when some expected values are not well-defined, we assume for simplicity that $\mathcal{X}$ is a bounded subset of $\mathbb{R}^d$ for som $d \geq 1$: $\mathcal{X} = \left\{x \in \mathbb{R}^d : \|x\|_2 \leq B_1\right\}$ for some $B_1 > 0$. Similarly, we assume that $\mathcal{Y}$ is a bounded subset of $\mathbb{R}$: $\mathcal{Y} = \{y \in \mathbb{R} : |y| \leq B_2\}$ for some $B_2 > 0$. We note that the construction of the regression-based IT will not require explicit knowledge of constants $B_1$ and $B_2$. First, we consider a setting where an instance either from the joint distribution or an instance from the product of the marginal distributions is observed at each round.

**Definition 3** (Proxy Setting). Suppose that we observe a stream of i.i.d. observations $((X_t, Y_t, W_t))_{t \geq 1}$, where $W_t \sim \mathrm{Rademacher}(1/2)$, the distribution of $(X_t, Y_t) \mid W_t = +1$ is $P_X \times P_Y$, and that of $(X_t, Y_t) \mid W_t = -1$ is $P_{XY}$. The goal is to design a test for the following pair of hypotheses:

$$
H_0 : P_{XY} = P_X \times P_Y, \tag{26a}
$$

$$
H_1 : P_{XY} \neq P_X \times P_Y. \tag{26b}
$$

**Oracle Proxy Sequential Regression-based IT.**   To construct an oracle test, we assume having access to the oracle predictor $g_\star : \mathcal{X} \to \mathcal{Y}$, e.g., the minimizer of the squared risk is $g_\star(x) = \mathbb{E}[Y \mid X = x]$. Formalizing the above intuition, we use $\mathbb{E}[W \ell(g_\star(X), Y)]$ as a natural way for measuring dependence between $X$ and $Y$. To enforce boundedness of the payoff functions, we use ideas of the tests for symmetry from [Ramdas et al., 2020, Shekhar and Ramdas, 2023, Podkopaev et al., 2023, Shaer et al., 2023], namely we use a composition with an odd function:

$$
f_\star^r(X_t, Y_t, W_t) = \tanh\left(s_\star \cdot W_t \cdot \ell(g_\star(X_t), Y_t)\right) \in [-1, 1], \tag{27}
$$

where $s_\star > 0$ is an appropriately selected scaling factor[4]. Since under $H_0$ in (26a), $s_\star \cdot W_t \cdot \ell(g_\star(X_t), Y_t)$ is a random variable that is symmetric around zero, it follows that $\mathbb{E}[f_\star^r(X_t, Y_t, W_t)] =$

---

[4]We note that rescaling is important for arguing about consistency and not the type I error control.

0, and, using the argument analogous to the proof of Theorem 1, we can easily deduce that a sequential IT based on $f_\star^{\mathrm{r}}$ controls the type I error control. The scaling factor $s_\star$ is selected in a way that guarantees that, if $H_1$ in (26b) is true and if $\mathbb{E}\left[W\ell(g_\star(X), Y)\right] > 0$, then $\mathbb{E}\left[f_\star^{\mathrm{r}}(X, Y, W)\right] > 0$, which is a sufficient condition for consistency of the oracle test. In particular, we show that it suffices to consider:

$$s_\star = \sqrt{\frac{2\mu_\star}{\nu_\star}}, \tag{28a}$$

$$\text{where} \quad \mu_\star = \mathbb{E}\left[W\ell(g_\star(X), Y)\right], \tag{28b}$$

$$\nu_\star = \mathbb{E}\left[(1+W)\left(\ell(g_\star(X), Y)\right)^3\right]. \tag{28c}$$

Without loss of generality, we assume that $\nu_\star$ is bounded away from zero (which is a very mild assumption since $\nu_\star$ essentially corresponds to a cubic risk of $g_\star$ on data drawn from the product of the marginal distributions $P_X \times P_Y$). Let the *oracle* regression-based wealth process $\left(\mathcal{K}_t^{\mathrm{r},\star}\right)_{t\geq 0}$ be defined by using the payoff function (27) with a scaling factor defined in (28a), along with a predictable sequence of betting fractions $(\lambda_t)_{t\geq 1}$ selected via the ONS strategy (Algorithm 1). We have the following result about the oracle regression-based IT, whose proof is deferred to Appendix D.4.

**Theorem 3.** *The following claims hold for the oracle sequential regression-based IT based on* $\left(\mathcal{K}_t^{\mathrm{r},\star}\right)_{t\geq 0}$:

1. *Suppose that $H_0$ in (26a) is true. Then the test ever stops with probability at most $\alpha$:* $\mathbb{P}_{H_1}\left(\tau < \infty\right) \leq \alpha.$

2. *Suppose that $H_1$ in (26b) is true. Further, suppose that: $\mathbb{E}\left[W\ell(g_\star(X), Y)\right] > 0$. Then the test is consistent: $\mathbb{P}_{H_1}\left(\tau < \infty\right) = 1.$*

**Practical Proxy Sequential Regression-based IT.** To construct a practical test, we use a sequence of predictors $(g_t)_{t\geq 1}$ that are updated sequentially as more data are observed. We write $\mathcal{A}_{\mathrm{r}} : \left(\cup_{t\geq 1}(\mathcal{X} \times \mathcal{Y})^t\right) \times \mathcal{G} \to \mathcal{G}$ to denote a chosen regressor learning algorithm which maps a training dataset of any size and previously used predictor, to an updated predictor. We start with $\mathcal{D}_0 = \emptyset$ and some initial guess $g_1 \in \mathcal{G}$. At round $t$, we use the payoff function:

$$f_t^{\mathrm{r}}(X_t, Y_t, W_t) = \tanh\left(s_t \cdot W_t \cdot \ell(g_t(X_t), Y_t)\right). \tag{29}$$

where a sequence of predictable scaling factors $(s_t)_{t\geq 1}$ is defined as follows: we set $s_0 = 0$ and define:

$$s_t = \sqrt{\frac{2\mu_t}{\nu_t}}, \tag{30a}$$

$$\text{where} \quad \mu_t = \left(\frac{1}{t-1}\sum_{i=1}^{t-1} W_i \cdot \ell(g_i(X_i), Y_i)\right) \vee 0, \tag{30b}$$

$$\nu_t = \frac{1}{t-1}\sum_{i=1}^{t-1}(1+W_i)\cdot\left(\ell(g_i(X_i), Y_i)\right)^3. \tag{30c}$$

After $(X_t, Y_t, W_t)$ has been used for betting, we update a training dataset: $\mathcal{D}_t = \mathcal{D}_{t-1} \cup \{(X_t, Y_t, W_t)\}$, and an existing predictor: $g_{t+1} = \mathcal{A}_{\mathrm{r}}(\mathcal{D}_t, g_t)$. We summarize this practical sequential 2ST in Algorithm 3.

For simplicity, we consider a class of functions $\mathcal{G} := \{g_\theta : \mathcal{X} \to \mathcal{Y}, \theta \in \Theta\}$ for some parameter set $\Theta$ which we assume to be a subset of a metric space. In this case, a sequence of predictors $(g_t)_{t\geq 1}$ is associated with the corresponding sequence of parameters $(\theta_t)_{t\geq 1}$: for $t \geq 1$, $g_t(\cdot) = g(\cdot; \theta_t)$ for some $\theta_t \in \Theta$. To argue about the consistency of the resulting test, we make two assumptions.

**Assumption 3** (Smoothness). We assume that:

- Predictors in $\mathcal{G}$ are $L_1$-Lipschitz smooth:

$$\sup_{x \in \mathcal{X}}|g(x;\theta) - g(x;\theta')| \leq L_1\|\theta - \theta'\|, \quad \forall \theta, \theta' \in \Theta. \tag{31}$$

---
**Algorithm 3** Proxy Sequential Regression-based IT
---
**Input:** significance level $\alpha \in (0,1)$, data stream $((X_t, Y_t, W_t))_{t \geq 1}$, $g_1(z) \equiv 0$, $\mathcal{A}_r$, $\mathcal{D}_0 = \emptyset$, $\lambda_1^{\text{ONS}} = 0$, $s_1 = 0$.
**for** $t = 1, 2, \ldots$ **do**
    Evaluate the payoff $f_t^r(X_t, Y_t, W_t)$ as in (29);
    Using $\lambda_t^{\text{ONS}}$, update the wealth process $\mathcal{K}_t^r$ as in (5);
    **if** $\mathcal{K}_t^r \geq 1/\alpha$ **then**
        Reject $H_0$ and stop;
    **else**
        Update the training dataset: $\mathcal{D}_t := \mathcal{D}_{t-1} \cup \{(X_t, Y_t)\}$;
        Update predictor: $g_{t+1} = \mathcal{A}_r(\mathcal{D}_t, g_t)$;
        Compute $s_{t+1}$ as in (30a);
        Compute $\lambda_{t+1}^{\text{ONS}}$ (Algorithm 1) using $f_t^r(X_t, Y_t, W_t)$;
---

- The loss function $\ell$ is $L_2$-Lipschitz smooth:

$$\sup_{\substack{x \in \mathcal{X} \\ y \in \mathcal{Y}}} |\ell(g(x; \theta), y) - \ell(g(x; \theta'), y)| \leq L_2 \sup_{x \in \mathcal{X}} |g(x; \theta) - g(x; \theta')|, \quad \forall \theta, \theta' \in \Theta. \quad (32)$$

In words, Assumption (31) states that the outputs of predictors, whose parameters are close, will also be close. Assumption (32) states that that the losses of two predictors, whose outputs are close, will also be close. For example, if $\mathcal{G}$ is a class of linear predictors: $g_\theta(x) = \theta^\top x$, $x \in \mathcal{X}$, then Assumption 3 will be trivially satisfied for the squared and the absolute losses if $\mathcal{X}$ and $\mathcal{Y}$ are bounded. Note that we do not need an explicit knowledge of $L_1$ or $L_2$ for designing a test. Second, we make a *learnability* assumption about algorithm $\mathcal{A}_r$.

**Assumption 4** (Learnability). Suppose that $H_1$ in (26b) is true. We assume that the regressor learning algorithm $\mathcal{A}_r$ is such that for the resulting sequence of parameters $(\theta_t)_{t \geq 1}$, it holds that $\theta_t \overset{\text{a.s.}}{\to} \theta_\star$, where $\theta_\star$ is a random variable taking values in $\Theta$ and $\mathbb{E}[W\ell(g(X; \theta_\star), Y) \mid \theta_\star] \overset{\text{a.s.}}{>} 0$, where $(X, Y, W) \perp\!\!\!\perp \theta_\star$.

We conclude with the following result for the practical proxy sequential regression-based IT, whose proof is deferred to Appendix D.4.

**Theorem 4.** *The following claims hold for the proxy sequential regression-based IT (Algorithm 3):*

1. *Suppose that $H_0$ in (26a) is true. Then the test ever stops with probability at most $\alpha$: $\mathbb{P}_{H_0}(\tau < \infty) \leq \alpha$.*

2. *Suppose that $H_1$ in (26b) is true. Further, suppose that Assumptions 3 and 4 are satisfied. Then the test is consistent: $\mathbb{P}_{H_1}(\tau < \infty) = 1$.*

**Sequential Regression-based Independence Test (Seq-R-IT).** Next, we instantiate this test for the sequential independence testing setting (as per Definition 2) where we observe sequence $((X_t, Y_t))_{t \geq 1}$, where $(X_t, Y_t) \overset{\text{iid}}{\sim} P_{XY}$, $t \geq 1$. Analogous to Section 2, we bet on the outcome of two observations drawn from the joint distribution $P_{XY}$. To proceed, we derandomize the payoff function (29) and consider

$$f_t^r((X_{2t-1}, Y_{2t-1}), (X_{2t}, Y_{2t})) = \frac{1}{4} \left( \tanh\left(s_t \cdot \ell\left(g_t(X_{2t-1}), Y_{2t}\right)\right) + \tanh\left(s_t \cdot \ell\left(g_t(X_{2t}), Y_{2t-1}\right)\right) \right)$$
$$- \frac{1}{4} \left( \tanh\left(s_t \cdot \ell\left(g_t(X_{2t}), Y_{2t}\right)\right) - \tanh\left(s_t \cdot \ell\left(g_t(X_{2t-1}), Y_{2t-1}\right)\right) \right). \quad (33)$$

After betting on the outcome of the $t$-th pair of observations from $P_{XY}$, we update a training dataset:

$$\mathcal{D}_t = \mathcal{D}_{t-1} \cup \{(X_{2t-1}, Y_{2t-1}), (X_{2t}, Y_{2t})\},$$

and a predictive model: $\hat{g}_{t+1} = \mathcal{A}_r(\mathcal{D}_t, \hat{g}_t)$.

## A.2   Synthetic Experiments

To evaluate the performance of Seq-R-IT, we consider the *Gaussian linear model*. Let $(X_t)_{t \geq 1}$ and $(\varepsilon_t)_{t \geq 1}$ denote two independent sequences of i.i.d. standard Gaussian random variables. For $t \geq 1$, we take

$$(X_t, Y_t) = (X_t, X_t \beta + \varepsilon_t),$$

where $\beta \neq 0$ implies nonzero linear correlation (hence dependence). We consider 20 values of $\beta$ equally spaced in $[0, 1/2]$. For the comparison, we use:

1. *Seq-R-IT with ridge regression.* We use ridge regression as an underlying model: $\hat{g}_t(x) = \beta_0^{(t)} + x\beta_1^{(t)}$, where

$$(\beta_0^{(t)}, \beta_1^{(t)}) = \arg\min_{\beta_0, \beta_1} \sum_{i=1}^{2(t-1)} (Y_i - X_i\beta_1 - \beta_0)^2 + \lambda\beta_1^2.$$

2. *Seq-C-IT with QDA.* Note that $P_{XY} = \mathcal{N}(\mu, \Sigma^+)$ and $P_X \times P_Y = \mathcal{N}(\mu, \Sigma^-)$, where

$$\mu = \begin{pmatrix} 0 \\ 0 \end{pmatrix}, \quad \Sigma^+ = \begin{pmatrix} 1 & \beta \\ \beta & 1 + \beta^2 \end{pmatrix}, \quad \Sigma^- = \begin{pmatrix} 1 & 0 \\ 0 & 1 + \beta^2 \end{pmatrix}.$$

For this problem, an oracle predictor which minimizes the misclassification risk is

$$g^\star(x, y) = \frac{\varphi((x,y); \mu^+, \Sigma^+) - \varphi((x,y); \mu^-, \Sigma^-)}{\varphi((x,y); \mu^-, \Sigma^-) + \varphi((x,y); \mu^+, \Sigma^+)} \in [-1, 1], \tag{34}$$

where $\varphi((x,y); \mu, \Sigma)$ denotes the density of the Gaussian distribution $\mathcal{N}(\mu, \Sigma)$ evaluated at $(x, y)$. Recall that $\mathcal{D}_{t-1} = \{(Z_i, +1)\}_{i \leq 2(t-1)} \cup \{(Z_i', -1)\}_{i \leq 2(t-1)}$ denotes the training dataset that is available at round $t$ for training a predictor $\hat{g}_t : \mathcal{X} \times \mathcal{Y} \to [-1, 1]$. We deploy Seq-C-IT with an estimator $\hat{g}_t$ of (34), obtained by using plug-in estimates of $\mu^+, \Sigma^+, \mu^-, \Sigma^-$, computed from $\mathcal{D}_{t-1}$:

$$\hat{\mu}_t^+ = \frac{1}{2(t-1)} \sum_{Z \in \mathcal{D}_{t-1}^+} Z, \qquad \hat{\Sigma}_t^+ = \left( \frac{1}{2(t-1)} \sum_{Z \in \mathcal{D}_{t-1}^+} ZZ^\top \right) - (\hat{\mu}_t^+)(\hat{\mu}_t^+)^\top,$$

and $\hat{\mu}_t^-, \hat{\Sigma}_t^-$ are computed similarly from $\mathcal{D}_t^-$.

In addition, we also include HSIC-based SKIT to the comparison and defer the details regarding kernel hyperparameters to Appendix E.1. We set the monitoring horizon to $T = 5000$ points from $P_{XY}$ and aggregate the results over 200 sequences of observations for each value of $\beta$. We illustrate the result in Figure 5: while Seq-R-IT has high power for large values of $\beta$, we observe its inferior performance against Seq-C-IT (and SKIT) under the harder settings. Improving regression-based betting strategies, e.g., designing better scaling factors that still yield a provably consistent test, is an open question for future research.

## B   Two-sample Testing with Unbalanced Classes

In Section 2, we developed a sequential 2ST under the assumption at each round, an instance from either $P$ or $Q$ is revealed with equal probability. Such assumption was reasonable for designing Seq-C-IT, where external randomization produced two instances from $P_{XY}$ and $P_X \times P_Y$ at each round. Next, we generalize our sequential 2ST to a more general setting of unbalanced classes.

**Definition 4** (Sequential two-sample testing with unbalanced classes). Let $\pi \in (0, 1)$. Suppose that we observe a stream of i.i.d. observations $((Z_t, W_t))_{t \geq 1}$, where $W_t \sim \text{Rademacher}(\pi)$, the distribution of $Z_t \mid W_t = +1$ is denoted $P$, and that of $Z_t \mid W_t = -1$ is denoted $Q$. We set the goal of designing a sequential test for the following pair of hypotheses:

$$H_0 : P = Q, \tag{35a}$$

$$H_1 : P \neq Q. \tag{35b}$$

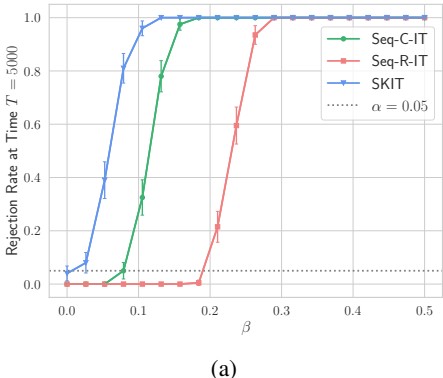

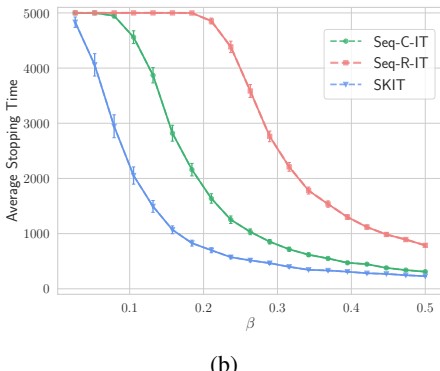

(a)

(b)

Figure 5: Comparison between Seq-R-IT, Seq-C-IT and HSIC-based SKIT under the Gaussian linear model. Inspecting Figure 5a at $\beta = 0$ confirms that all tests control the type I error. Non-surprisingly, kernel-based SKIT performs better than predictive tests under this model (no localized dependence). We also observe that Seq-C-IT performs better than Seq-R-IT.

For what follows, we will focus on the payoff based on the squared risk due to its relationship to the likelihood-ratio-based test (Remark 3). In particular, after correcting the likelihood under the null in (20) to account for a general positive class proportion $\pi$, we can deduce that (see Appendix D.5):

$$(1-\lambda_t)\cdot 1 + \lambda_t \cdot \frac{(\eta_t(Z_t))^{\mathbb{1}\{W_t=1\}}(1-\eta_t(Z_t))^{\mathbb{1}\{W_t=0\}}}{(\pi)^{\mathbb{1}\{W_t=1\}}(1-\pi)^{\mathbb{1}\{W_t=0\}}} = 1 + \lambda_t \cdot \frac{W_t(g_t(Z_t)-(2\pi-1))}{1+W_t(2\pi-1)}, \quad (36)$$

where $\eta_t(z) = (g_t(z)+1)/2$, and hence, a natural payoff function for the case with unbalanced classes is

$$f_t^{\mathrm{u}}(Z_t, W_t) = \frac{W_t(g_t(Z_t)-(2\pi-1))}{1+W_t(2\pi-1)}. \quad (37)$$

Note that the payoff for the balanced case (22b) is recovered by setting $\pi = 1/2$. It is easy to check that (see Appendix D.5): (a) $f_t^{\mathrm{u}}(z,w) \geq -1$ for any $(z,w) \in \mathcal{Z} \times \{-1,1\}$, and (b) if $H_0$ in (35a) is true, then $\mathbb{E}_{H_0}[f_t^{\mathrm{u}}(Z_t, W_t) \mid \mathcal{F}_{t-1}] = 0$, where $\mathcal{F}_{t-1} = \sigma(\{(Z_i, W_i)\}_{i \leq t-1})$. This in turn implies that a wealth process that relies on the payoff function $f_t^{\mathrm{u}}$ in (37) is a nonnegative martingale, and hence, the corresponding sequential 2ST is valid. However, the positive class proportion $\pi$, needed to use the payoff function (37), is generally unknown beforehand. First, let us consider the case when $\lambda_t = 1, t \geq 1$. In this case, the wealth of a gambler that uses the payoff function (37) after round $t$ is

$$\mathcal{K}_t = \frac{\prod_{i=1}^{t}(\eta_i(Z_i))^{\mathbb{1}\{W_i=1\}}(1-\eta_i(Z_i))^{\mathbb{1}\{W_i=0\}}}{\prod_{i=1}^{t}\pi^{\mathbb{1}\{W_i=1\}}(1-\pi)^{\mathbb{1}\{W_i=0\}}}. \quad (38)$$

Note that:

$$\hat{\pi}_t := \frac{1}{t}\sum_{i=1}^{t}\mathbb{1}\{W_t=1\} = \arg\max_{\pi \in [0,1]}\left(\prod_{i=1}^{t}\pi^{\mathbb{1}\{W_i=1\}}(1-\pi)^{\mathbb{1}\{W_i=0\}}\right),$$

is the MLE for $\pi$ computed from $\{W_i\}_{i \leq t}$. In particular, if we consider a process $(\tilde{\mathcal{K}}_t)_{t \geq 0}$, where

$$\tilde{\mathcal{K}}_t := \frac{\prod_{i=1}^{t}(\eta_i(Z_i))^{\mathbb{1}\{W_i=1\}}(1-\eta_i(Z_i))^{\mathbb{1}\{W_i=0\}}}{\prod_{i=1}^{t}(\hat{\pi}_t)^{\mathbb{1}\{W_i=1\}}(1-\hat{\pi}_t)^{\mathbb{1}\{W_i=0\}}}, \quad t \geq 1,$$

it follows that $\tilde{\mathcal{K}}_t \leq \mathcal{K}_t, \forall t \geq 1$, meaning that $(\tilde{\mathcal{K}}_t)_{t \geq 0}$ is a process that is upper bounded by a nonnegative martingale with initial value one. This in turn implies that a test based on $(\tilde{\mathcal{K}}_t)_{t \geq 0}$ is a valid level-$\alpha$ sequential 2ST for the case of unknown class proportions. This idea underlies the running MLE sequential likelihood ratio test of Wasserman et al. [2020] and has been recently considered in the context of two-sample testing by Pandeva et al. [2022]. In case of nontrivial betting fractions: $(\lambda_t)_{t \geq 1}$, representation of the wealth process (38) no longer holds, and to proceed, we modify the rules of the game and use minibatching. A bet is placed on every $b$ (say, 5 or 10) observations, meaning

that for a given minibatch size $b \geq 1$, at round $t$ we bet on $\{(Z_{b(t-1)+i}, W_{b(t-1)+i})\}_{i \in \{1,\dots,b\}}$. The MLE of $\pi$ computed from the $t$-th minibatch is

$$\hat{\pi}_t = \frac{1}{b} \sum_{i=b(t-1)+1}^{bt} \mathbb{1}\{W_i = +1\}.$$

We consider a payoff function of the following form:

$$f_t^{\mathrm{u}}\left(\{(Z_{b(t-1)+i}, W_{b(t-1)+i})\}_{i \in \{1,\dots,b\}}\right) = \prod_{i=b(t-1)+1}^{bt} \left(\frac{1 + W_i g_t(Z_i)}{1 + W_i(2\hat{\pi}_t - 1)}\right) - 1. \quad (39)$$

In words, the above payoff essentially compares the performance of a predictor $g_t$, trained on $\{(Z_i, W_i)\}_{i \leq b(t-1)}$ and evaluated on the $t$-th minibatch, to that of a trivial baseline predictor to form a bet. In particular, setting $b = 1$ yields a valid, yet a powerless test. Indeed, we have $\hat{\pi}_t = \mathbb{1}\{W_t = 1\} = (W_t + 1)/2$. In this case, the payoff (39) reduces to

$$\frac{W_t(g_t(Z_t) - (2\hat{\pi}_t - 1))}{1 + W_t(2\hat{\pi}_t - 1)} = \frac{W_t g_t(Z_t) - 1}{2} \overset{\text{a.s.}}{\in} [-1, 0],$$

implying that the wealth can not grow even if the null is false. Define a wealth processes $(\mathcal{K}_t^{\mathrm{u}})_{t \geq 0}$ based on the payoff functions (39) along with a predictable sequence of betting fractions $(\lambda_t)_{t \geq 1}$ selected via ONS strategy (Algorithm 1). Let $\mathcal{F}_t = \sigma(\{(Z_i, W_i)\}_{i \leq bt})$ for $t \geq 1$, with $\mathcal{F}_0$ denoting a trivial sigma-algebra. We conclude with the following result, whose proof is deferred to Appendix D.5.

**Theorem 5.** *Suppose that $H_0$ in (35a) is true. Then $(\mathcal{K}_t^{\mathrm{u}})_{t \geq 0}$ is a nonnegative supermartingale adapted to $(\mathcal{F}_t)_{t \geq 0}$. Hence, the sequential 2ST based on $(\mathcal{K}_t^{\mathrm{u}})_{t \geq 0}$ satisfies: $\mathbb{P}_{H_0}(\tau < \infty) \leq \alpha$.*

## C  Testing under Distribution Drift

First, we define the problem of two-sample testing when at each round instances from both distributions are observed.

**Definition 5** (Sequential two-sample testing). Suppose that we observe that a stream of observations: $((X_t, Y_t))_{t \geq 1}$, where $(X_t, Y_t) \overset{\text{iid}}{\sim} P_X \times P_Y$ for $t \geq 1$. The goal is to design a sequential test for

$$H_0 : (X_t, Y_t) \overset{\text{iid}}{\sim} P_X \times P_Y \text{ and } P_X = P_Y, \quad (40a)$$

$$H_1 : (X_t, Y_t) \overset{\text{iid}}{\sim} P_X \times P_Y \text{ and } P_X \neq P_Y. \quad (40b)$$

Under the two-sample testing setting (Definition 5), we label observations from $P_Y$ as positive ($+1$) and observations from $P_X$ as negative ($-1$). We write $\mathcal{A}_c^{\mathrm{2ST}} : (\cup_{t \geq 1}(\mathcal{X} \times \{-1, +1\})^t) \times \mathcal{G} \to \mathcal{G}$ to denote a chosen learning algorithm which maps a training dataset of any size and previously used predictor, to an updated predictor. We start with $\mathcal{D}_0 = \emptyset$ and $g_1 : g_1(x) = 0, \forall x \in \mathcal{X}$. At round $t$, we bet using derandomized versions of the payoffs (22), namely

$$f_t^{\mathrm{m}}(X_t, Y_t) = \tfrac{1}{2}\left(\operatorname{sign}[g_t(Y_t)] - \operatorname{sign}[g_t(X_t)]\right), \quad (41a)$$

$$f_t^{\mathrm{s}}(X_t, Y_t) = \tfrac{1}{2}\left(g_t(Y_t) - g_t(X_t)\right). \quad (41b)$$

After $(X_t, Y_t)$ has been used for betting, we update a training dataset and an existing predictor:

$$\mathcal{D}_t = \mathcal{D}_{t-1} \cup \{(Y_t, +1), (X_t, -1)\}, \quad g_{t+1} = \mathcal{A}_c^{\mathrm{2ST}}(\mathcal{D}_t, g_t).$$

**Testing under Distribution Drift.** Batch two-sample and independence tests generally rely on either a cutoff computed using the asymptotic null distribution of a chosen test statistic (if tractable) or a permutation p-value. Both approaches require imposing i.i.d. (or exchangeability, for the latter option) assumption about the data distribution, and if the distribution drifts, both approaches fail to guarantee the type I error control. In contrast, Seq-C-2ST and Seq-C-IT remain valid beyond the i.i.d. setting by construction (analogous to tests developed in [Shekhar and Ramdas, 2023, Podkopaev et al., 2023]). First, we define the problems of sequential two-sample and independence testing under distribution drift.

**Definition 6** (Sequential two-sample testing under distribution drift). Suppose that we observe that a stream of independent observations: $((X_t, Y_t))_{t \geq 1}$, where $(X_t, Y_t) \sim P_X^{(t)} \times P_Y^{(t)}$, $t \geq 1$. The goal is to design a sequential test for the following pair of hypotheses:

$$H_0 : P_X^{(t)} = P_Y^{(t)}, \; \forall t, \tag{42a}$$

$$H_1 : \exists t' : P_X^{(t')} \neq P_Y^{(t')}. \tag{42b}$$

**Definition 7** (Sequential independence testing under distribution drift). Suppose that we observe that a stream of independent observations from the joint distribution which drifts over time: $((X_t, Y_t))_{t \geq 1}$, where $(X_t, Y_t) \sim P_{XY}^{(t)}$. The goal is to design a sequential test for the following pair of hypotheses:

$$H_0 : P_{XY}^{(t)} = P_X^{(t)} \times P_Y^{(t)}, \; \forall t, \tag{43a}$$

$$H_1 : \exists t' : P_{XY}^{(t')} \neq P_X^{(t')} \times P_Y^{(t')}. \tag{43b}$$

The superscripts highlight that, in contrast to the standard i.i.d. setting (Definitions 5 and 2), the underlying distributions may drift over time. For independence testing, we need to impose an additional assumption that enables reasoning about the type I error control of Seq-C-IT.

**Assumption 5.** Consider the setting of independence testing under distribution drift (Definition 7). We assume that for each $t \geq 1$, it holds that either $P_X^{(t-1)} = P_X^{(t)}$ or $P_Y^{(t-1)} = P_Y^{(t)}$, meaning that at each step either the distribution of $X$ changes or that of $Y$ changes, but not both simultaneously[5].

We have the following result about the type I error control of our tests under distribution drift.

**Corollary 2.** *The following claims hold:*

1. *Suppose that $H_0$ in (42a) is true. Then Seq-C-2ST satisfies:* $\mathbb{P}_{H_0}(\tau < \infty) \leq \alpha$.

2. *Suppose that $H_0$ in (43a) is true. Further, suppose that Assumption 5 is satisfied. Then Seq-C-IT satisfies:* $\mathbb{P}_{H_0}(\tau < \infty) \leq \alpha$.

The above result follows from the fact the payoff functions underlying Seq-C-2ST (41) and Seq-C-IT (23) are valid under the more general null hypotheses (42a) and (43a) respectively. The rest of the proof of Corollary 2 follows the same steps as that of Theorem 2, and we omit the details. We conclude with an example which shows that Assumption 5 is necessary for the type I error control.

**Example 2.** Consider the following case when the null $H_0$ in (43a) is true, but Assumption 5 is not satisfied. We show that Seq-C-IT fails to control type I error (at any prespecified level $\alpha \in (0,1)$), and for simplicity, focus on the payoff function based on the squared risk (23). Suppose that we observe a sequence of observations: $((X_t, Y_t))_{t \geq 1}$, where $(X_t, Y_t) = (t + W_t, t + V_t)$ and $W_t, V_t \overset{\text{iid}}{\sim} \text{Bern}(1/2)$. It suffices to show that there exists a sequence of predictors $(g_t)_{t \geq 1}$, for which

$$\liminf_{t \to \infty} \frac{1}{t} \sum_{i=1}^{t} f_t^s((X_{2t-1}, Y_{2t-1}), (X_{2t}, Y_{2t})) \overset{\text{a.s.}}{>} 0. \tag{44}$$

If (44) holds, then using the same argument as in the proof of Theorem 2, one can then deduce that $\mathbb{P}(\tau < \infty) = 1$. Consider the following sequence of predictors $(g_t)_{t \geq 1}$:

$$g_t(x, y) = \left(\left(x - \left(2t - \tfrac{1}{2}\right)\right)\left(y - \left(2t - \tfrac{1}{2}\right)\right) \wedge 1\right) \vee -1.$$

We have:

$$g_t(X_{2t}, Y_{2t}) = \left(\left(W_{2t} + \tfrac{1}{2}\right)\left(V_{2t} + \tfrac{1}{2}\right) \wedge 1\right) \vee -1,$$

$$g_t(X_{2t-1}, Y_{2t-1}) = \left(W_{2t-1} - \tfrac{1}{2}\right)\left(V_{2t-1} - \tfrac{1}{2}\right),$$

$$g_t(X_{2t}, Y_{2t-1}) = \left(W_{2t} + \tfrac{1}{2}\right)\left(V_{2t-1} - \tfrac{1}{2}\right),$$

$$g_t(X_{2t-1}, Y_{2t}) = \left(W_{2t-1} - \tfrac{1}{2}\right)\left(V_{2t} + \tfrac{1}{2}\right).$$

Simple calculation shows that:

$$\mathbb{E}\left[g_t(X_{2t}, Y_{2t})\right] = 11/16, \quad \mathbb{E}\left[g_t(X_{2t-1}, Y_{2t-1})\right] = \mathbb{E}\left[g_t(X_{2t}, Y_{2t-1})\right] = \mathbb{E}\left[g_t(X_{2t-1}, Y_{2t})\right] = 0$$

and hence, for all $t \geq 1$, it holds that $\mathbb{E}\left[f_t^s((X_{2t-1}, Y_{2t-1}), (X_{2t}, Y_{2t}))\right] = 11/64 > 0$. This in turn implies (44), and hence, we conclude that Seq-C-IT fails to control the type I error.

---

[5]Technically, a slightly weaker condition suffices — at odd $t$, the distribution can change arbitrarily, but at even $t$, either the distribution of $X$ changes or that of $Y$ changes but not both; however, this weaker condition is slightly less intuitive than the stated condition.

## C.1 Synthetic Experiments

Under the alternative, the power of our test depends on the performance of a classifier according to misclassification/squared risk. If the distribution drifts over time, then models updated via online gradient descent may retain predictive power. In such cases, our test will still have high power despite the present distribution drift. Of course, if the distribution shifts adversarially, the method will not have power, but neither will any other method. So the implicit goal is to maintain type-1 error control under the null despite shifting distributions (beyond the i.i.d. assumption typically made in the literature) while retaining power against reasonable (but not all) distribution drifts. To put this into context, we consider four settings:

1. $P = Q = \mathcal{N}(0, 1)$, i.e., the 2ST null is true;

2. $P = \mathcal{N}(0, 1)$ and $Q = \mathcal{N}(0.5, 1)$, i.e., the 2ST null is false;

3. Up to time $t = 250$, $P^{(t)} = Q^{(t)} = \mathcal{N}(0, 1)$, but for $t > 251$, we have $P^{(t)} = \mathcal{N}(0, 1)$ and $Q^{(t)} = \mathcal{N}(0.5, 1)$, i.e., there is a distribution shift and the 2ST null is false;

4. Up to time $t = 250$, $P^{(t)} = Q^{(t)} = \mathcal{N}(0, 1)$, but for $t > 251$, $P^{(t)} = Q^{(t)} = \mathcal{N}(0.5, 1)$, i.e., there is a change in distribution yet the 2ST null is still true.

We use a standard logistic regression model as an underlying predictor with weights updated via online gradient descent and monitor the tests until $T = 2000$ observations. In Figure 6, we observe that our test controls the type-1 error whenever the 2ST null is true even if there is a shift in distribution, and retains high power if the alternative is true.

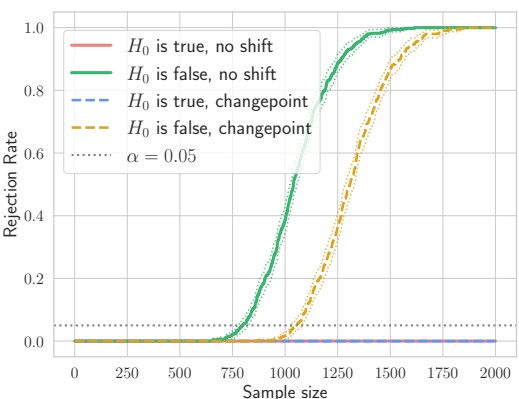

Figure 6: Average rejection rates for four settings: solid lines correspond to the standard i.i.d. setting and dashed lines correspond to the settings that involve distribution shift. Despite there being a shift in distribution, our test that utilizes logistic regression learned via online gradient descent controls the type-1 error under a more general 2ST null hypothesis and has power under the alternative. Results are aggregated over 500 random sequences from the corresponding distributions.

# D Proofs

## D.1 Auxiliary Results

**Proposition 2** (Ville's inequality [Ville, 1939]). *Suppose that $(\mathcal{M}_t)_{t \geq 0}$ is a nonnegative supermartingale process adapted to a filtration $(\mathcal{F}_t)_{t \geq 0}$. Then, for any $a > 0$ it holds that:*

$$\mathbb{P}\left(\exists t \geq 1 : \mathcal{M}_t \geq a\right) \leq \frac{\mathbb{E}\left[\mathcal{M}_0\right]}{a}.$$

## D.2 Supporting Lemmas

**Lemma 6.** *Consider sequential two-sample testing setting (Definition 1). Suppose that a predictor $g \in \mathcal{G}$ satisfies $\mathbb{E}\left[f(Z, W)\right] > 0$, where $f(z, w) := wg(z)$.*

(a) *Consider the wealth process $(\mathcal{K}_t)_{t\geq 0}$ based on $f$ along with the ONS strategy for selecting betting fractions (Algorithm 1). Then we have the following lower bound on the growth rate of the wealth process:*

$$\liminf_{t\to\infty} \frac{\log \mathcal{K}_t}{t} \overset{\text{a.s.}}{\geq} \frac{1}{4}\left(\frac{(\mathbb{E}\left[f(Z,W)\right])^2}{\mathbb{E}\left[f^2(Z,W)\right]} \wedge \mathbb{E}\left[f(Z,W)\right]\right). \tag{45}$$

(b) *For $\lambda_\star = \arg\max_{\lambda\in[-0.5,0.5]} \mathbb{E}\left[\log(1+\lambda f(Z,W))\right]$, it holds that:*

$$\mathbb{E}\left[\log(1+\lambda_\star f(Z,W))\right] \leq \frac{4}{3}\cdot\frac{(\mathbb{E}\left[f(Z,W)\right])^2}{\mathbb{E}\left[(f(Z,W))^2\right]} \wedge \frac{\mathbb{E}\left[f(Z,W)\right]}{2}. \tag{46}$$

*Analogous result holds when the payoff function $f(z,w) := w\cdot\text{sign}\left[g(z)\right]$ is used instead.*

*Proof.*     (a) Under the ONS betting strategy, for any sequence of outcomes $(f_t)_{t\geq 1}$, $f_t \in [-1,1]$, it holds that (see the proof of Theorem 1 in [Cutkosky and Orabona, 2018]):

$$\log \mathcal{K}_t(\lambda_0) - \log \mathcal{K}_t = O\left(\log\left(\sum_{i=1}^t f_i^2\right)\right), \tag{47}$$

where $\mathcal{K}_t(\lambda_0)$ is the wealth of any constant betting strategy $\lambda_0 \in [-1/2,1/2]$ and $\mathcal{K}_t$ is the wealth corresponding to the ONS betting strategy. Hence, it follows that

$$\frac{\log \mathcal{K}_t}{t} \geq \frac{\log \mathcal{K}_t(\lambda_0)}{t} - C\cdot\frac{\log t}{t}, \tag{48}$$

for some absolute constant $C > 0$. Next, consider

$$\lambda_0 = \frac{1}{2}\left(\left(\frac{\sum_{i=1}^t f_i}{\sum_{i=1}^t f_i^2} \wedge 1\right)\vee 0\right).$$

We obtain:

$$\frac{\log \mathcal{K}_t(\lambda_0)}{t} = \frac{1}{t}\sum_{i=1}^t \log(1+\lambda_0 f_i)$$

$$\overset{\text{(a)}}{\geq} \frac{1}{t}\sum_{i=1}^t(\lambda_0 f_i - \lambda_0^2 f_i^2) \tag{49}$$

$$= \left(\frac{\frac{1}{t}\sum_{i=1}^t f_i}{4}\vee 0\right)\cdot\left(\frac{\frac{1}{t}\sum_{i=1}^t f_i}{\frac{1}{t}\sum_{i=1}^t f_i^2}\wedge 1\right),$$

where in (a) we used that $\log(1+x) \geq x - x^2$ for $x \in [-1/2,1/2]$. From (48), it then follows that:

$$\liminf_{t\to\infty}\frac{\log \mathcal{K}_t}{t} \overset{\text{a.s.}}{\geq} \left(\frac{\mathbb{E}\left[f(Z,W)\right]}{4}\vee 0\right)\cdot\left(\frac{\mathbb{E}\left[f(Z,W)\right]}{\mathbb{E}\left[f^2(Z,W)\right]}\wedge 1\right)$$

$$= \frac{1}{4}\left(\frac{(\mathbb{E}\left[f(Z,W)\right])^2}{\mathbb{E}\left[f^2(Z,W)\right]}\wedge \mathbb{E}\left[f(Z,W)\right]\right),$$

which completes the proof of the first assertion of the lemma.

(b) Since $\log(1+x) \leq x - 3x^2/8$ for any $x \in [-0.5,0.5]$, we know that:

$$\mathbb{E}\left[\log\left(1+\lambda_\star f(Z,W)\right)\right] \leq \mathbb{E}\left[\lambda_\star f(Z,W) - \frac{3}{8}\left(\lambda_\star f(Z,W)\right)^2\right]$$

$$\leq \max_{\lambda\in[-0.5,0.5]}\left(\lambda\cdot\mathbb{E}\left[f(Z,W)\right] - \frac{3\lambda^2}{8}\cdot\mathbb{E}\left[(f(Z,W))^2\right]\right).$$

The optimizer of the above is

$$\tilde{\lambda} = \frac{4\mathbb{E}\left[f(Z,W)\right]}{3\mathbb{E}\left[(f(Z,W))^2\right]} \wedge \frac{1}{2}.$$

Hence, as long as $\mathbb{E}\left[f(Z,W)\right] \leq (3/8) \cdot \mathbb{E}\left[(f(Z,W))^2\right]$, we have:

$$\mathbb{E}\left[\log\left(1 + \lambda_\star f(Z,W)\right)\right] \leq \frac{2}{3} \frac{(\mathbb{E}\left[f(Z,W)\right])^2}{\mathbb{E}\left[(f(Z,W))^2\right]}. \tag{50}$$

If however, $\mathbb{E}\left[f(Z,W)\right] > (3/8) \cdot \mathbb{E}\left[(f(Z,W))^2\right]$, then we know that:

$$\mathbb{E}\left[\log\left(1 + \lambda_\star f(Z,W)\right)\right] \leq \frac{\mathbb{E}\left[f(Z,W)\right]}{2}.$$

To bring it to a convenient form, we multiply the upper bound in (50) by two and get the bound (46), which completes the proof of the second assertion of the lemma.

$\square$

### D.3 Proofs for Section 2

**Proposition 1.** *Fix an arbitrary predictor $g \in \mathcal{G}$. The following claims hold[6]:*

1. *For the misclassification risk, we have that:*

$$\sup_{s \in [0,1]} \left(\tfrac{1}{2} - R_{\mathrm{m}}(sg)\right) = \left(\tfrac{1}{2} - R_{\mathrm{m}}(g)\right) \vee 0 = \left(\tfrac{1}{2} \cdot \mathbb{E}\left[W \cdot \mathrm{sign}\left[g(Z)\right]\right]\right) \vee 0. \tag{9}$$

2. *For the squared risk, we have that:*

$$\sup_{s \in [0,1]} \left(1 - R_{\mathrm{s}}(sg)\right) \geq \left(\mathbb{E}\left[W \cdot g(Z)\right] \vee 0\right) \cdot \left(\frac{\mathbb{E}\left[W \cdot g(Z)\right]}{\mathbb{E}\left[g^2(Z)\right]} \wedge 1\right) \tag{10}$$

*Further, $d_{\mathrm{s}}(P,Q) > 0$ if and only if there exists $g \in \mathcal{G}$ such that $\mathbb{E}\left[W \cdot g(Z)\right] > 0$.*

*Proof.*      1. The first equality in (9) follows from two facts: (a) for any $g \in \mathcal{G}$ and any $s \in (0,1]$, it holds that $R_{\mathrm{m}}(sg) = R_{\mathrm{m}}(g)$, (b) $R_{\mathrm{m}}(0) = 1/2$. The second equality easily follows from the following fact: $\mathrm{sign}\left[x\right]/2 = 1/2 - \mathbb{1}\left\{x < 0\right\}$.

2. Consider an arbitrary predictor $g \in \mathcal{G}$. Let us consider all possible scenarios:

    (a) If $\mathbb{E}\left[W \cdot g(Z)\right] \leq 0$, then the RHS of (10) is zero. For the LHS of (10), we have that:

$$\sup_{s \in [0,1]} \left(1 - R_{\mathrm{s}}(sg)\right) \geq 1 - R_{\mathrm{s}}(0) = 0,$$

    so the bound (10) holds.

    (b) Next, assume that $\mathbb{E}\left[W \cdot g(Z)\right] > 0$, then it is easy to derive that:

$$s_\star := \arg\max_{s \in [0,1]} \left(1 - R_{\mathrm{s}}(sg)\right) = \frac{\mathbb{E}\left[W \cdot g(Z)\right]}{\mathbb{E}\left[g^2(Z)\right]} \wedge 1. \tag{51}$$

    A simple calculation shows that:

$$1 - R_{\mathrm{s}}(s_\star g) \geq \mathbb{E}\left[W \cdot g(Z)\right] \cdot \left(\frac{\mathbb{E}\left[W \cdot g(Z)\right]}{\mathbb{E}\left[g^2(Z)\right]} \wedge 1\right),$$

    and hence, we conclude that the bound (10) holds.

---

[6]$\vee$ and $\wedge$ denote a maximum and a minimum respectively: $a \vee b = \max\{a,b\}$, $a \wedge b = \min\{a,b\}$.

To establish the second part of the statement, note that $d_s(P,Q) > 0$ iff there is a predictor $g \in \mathcal{G}$ such that $R_s(g) < 1$. For the squared risk, we have:

$$1 - R_s(g) = 2\mathbb{E}\left[W \cdot g(Z)\right] - \mathbb{E}\left[g^2(Z)\right], \tag{52}$$

and hence, $R_s(g) < 1$ trivially implies that $\mathbb{E}\left[W \cdot g(Z)\right] > 0$. The converse implication trivially follows from (10). Hence, the result follows.

$\square$

**Theorem 1.** *The following claims hold:*

1. *Suppose that $H_0$ in (1a) is true. Then the oracle sequential test based on either $(\mathcal{K}_t^{m,\star})_{t\geq 0}$ or $(\mathcal{K}_t^{s,\star})_{t\geq 0}$ ever stops with probability at most $\alpha$: $\mathbb{P}_{H_0}(\tau < \infty) \leq \alpha$.*

2. *Suppose that $H_1$ in (1b) is true. Then:*

   (a) *The growth rate of the oracle wealth process $(\mathcal{K}_t^{m,\star})_{t\geq 0}$ satisfies:*

   $$\liminf_{t\to\infty} \left(\tfrac{1}{t}\log \mathcal{K}_t^{m,\star}\right) \overset{a.s.}{\geq} \left(\tfrac{1}{2} - R_m(g_\star)\right)^2. \tag{14}$$

   *If $R_m(g_\star) < 1/2$, then the test based on $(\mathcal{K}_t^{m,\star})_{t\geq 0}$ is consistent: $\mathbb{P}_{H_1}(\tau < \infty) = 1$. Further, the optimal growth rate achieved by $\lambda_\star^m$ in (13) satisfies:*

   $$\mathbb{E}\left[\log(1 + \lambda_\star^m f_\star^m(Z,W))\right] \leq \left(\tfrac{16}{3} \cdot \left(\tfrac{1}{2} - R_m(g_\star)\right)^2 \wedge \left(\tfrac{1}{2} - R_m(g_\star)\right)\right). \tag{15}$$

   (b) *The growth rate of the oracle wealth process $(\mathcal{K}_t^{s,\star})_{t\geq 0}$ satisfies:*

   $$\liminf_{t\to\infty} \left(\tfrac{1}{t}\log \mathcal{K}_t^{s,\star}\right) \overset{a.s.}{\geq} \tfrac{1}{4} \cdot \mathbb{E}\left[W \cdot g_\star(Z)\right]. \tag{16}$$

   *If $\mathbb{E}\left[W \cdot g_\star(Z)\right] > 0$, then the test based on $(\mathcal{K}_t^{s,\star})_{t\geq 0}$ is consistent: $\mathbb{P}_{H_1}(\tau < \infty) = 1$. Further, the optimal growth rate achieved by $\lambda_\star^s$ in (13) satisfies:*

   $$\mathbb{E}\left[\log(1 + \lambda_\star^s f_\star^s(Z,W))\right] \leq \tfrac{1}{2} \cdot \mathbb{E}\left[W \cdot g_\star(Z)\right]. \tag{17}$$

*Proof.* 1. We trivially have that the payoff functions (11a) and (11b) are bounded: $\forall (z,w) \in \mathcal{Z} \times \{-1,1\}$, it holds that $f_\star^m(z,w) \in [-1,1]$ and $f_\star^s(z,w) \in [-1,1]$. Further, under the null $H_0$ in (1a), it trivially holds that $\mathbb{E}_{H_0}\left[f_\star^m(Z_t, W_t) \mid \mathcal{F}_{t-1}\right] = \mathbb{E}_{H_0}\left[f_\star^s(Z_t, W_t) \mid \mathcal{F}_{t-1}\right] = 0$, where $\mathcal{F}_{t-1} = \sigma(\{(Z_i, W_i)\}_{i \leq t-1})$. Since ONS betting fractions $\left(\lambda_t^{\mathrm{ONS}}\right)_{t\geq 1}$ are predictable, we conclude that the resulting wealth process is a nonnegative martingale. The assertion of the Theorem then follows directly from Ville's inequality (Proposition 2) when $a = 1/\alpha$.

2. Suppose that $H_1$ in (1b) is true. First, we prove the results for the lower bounds:

   (a) Consider the wealth process based on the misclassification risk $(\mathcal{K}_t^{m,\star})_{t\geq 0}$. Note that for all $t \geq 1$:

   $$\mathbb{E}\left[f_\star^m(Z_t, W_t)\right] = 2 \cdot \left(\frac{1}{2} - R_m(g_\star)\right), \quad (f_\star^m(Z_t, W_t))^2 = 1.$$

   Since $\mathbb{E}\left[f_\star^m(Z_t, W_t)\right] \in [0,1]$, we also have $(\mathbb{E}\left[f_\star^m(Z_t, W_t)\right])^2 \leq \mathbb{E}\left[f_\star^m(Z_t, W_t)\right]$. From the first part of Lemma 6, it follows that:

   $$\liminf_{t\to\infty} \frac{\log \mathcal{K}_t^{m,\star}}{t} \overset{a.s.}{\geq} \frac{1}{4}\left(\mathbb{E}\left[f_\star^m(Z_t, W_t)\right]\right)^2 = \left(\frac{1}{2} - R_m(g_\star)\right)^2.$$

   From the second part of Lemma 6, and (46) in particular, it follows that:

   $$\mathbb{E}\left[\log\left(1 + \lambda_\star^m f_\star^m(Z,W)\right)\right] \leq \left(\frac{16}{3} \cdot \left(\frac{1}{2} - R_m(g_\star)\right)^2 \wedge \left(\frac{1}{2} - R_m(g_\star)\right)\right).$$

   The first term in the above is smaller or equal than the second one whenever $R_m(g_\star) \geq 5/16$. We conclude that the assertion of the theorem is true.

(b) Next, we consider the wealth process based on the squared error: $(\mathcal{K}_t^{s,\star})_{t\geq0}$. Note that:

$$\mathbb{E}\left[f_\star^s(Z_t, W_t)\right] = \mathbb{E}\left[W \cdot g_\star(Z)\right],$$

$$\mathbb{E}\left[(f_\star^s(Z_t, W_t))^2\right] = \mathbb{E}\left[g_\star^2(Z)\right],$$

and hence from Lemma 6, it follows that:

$$\liminf_{t\to\infty} \frac{\log \mathcal{K}_t^{s,\star}}{t} \overset{\text{a.s.}}{\geq} \frac{1}{4}\left(\frac{(\mathbb{E}\left[W \cdot g_\star(Z)\right])^2}{\mathbb{E}\left[g_\star^2(Z)\right]} \wedge \mathbb{E}\left[W \cdot g_\star(Z)\right]\right). \tag{53}$$

In the above, we assume that the following case is not possible: $g_\star(Z) \overset{\text{a.s.}}{=} 0$ (for such $g_\star$, the corresponding expected margin and the growth rate of the resulting wealth process are clearly zero, and will still be highlighted in our resulting bound). Next, note that since $g_\star \in \arg\min_{g\in\mathcal{G}} R_s(g)$, we have that:

$$1 - R_s(g_\star) = \sup_{s\in[0,1]}(1 - R_s(sg_\star)),$$

meaning that $g_\star$ can not be improved by scaling with $s < 1$. From Proposition 1, and (51) in particular, it follows that:

$$\frac{\mathbb{E}\left[W \cdot g_\star(Z)\right]}{\mathbb{E}\left[g_\star^2(Z)\right]} \geq 1, \tag{54}$$

and hence, the bound (53) reduces to

$$\liminf_{t\to\infty} \frac{\log \mathcal{K}_t^{s,\star}}{t} \overset{\text{a.s.}}{\geq} \frac{\mathbb{E}\left[W \cdot g_\star(Z)\right]}{4}.$$

From the second part of Lemma 6, it follows that:

$$\mathbb{E}\left[\log\left(1 + \lambda_\star^s f_\star^s(Z, W)\right)\right] \leq \frac{4}{3}\frac{(\mathbb{E}\left[W \cdot g_\star(Z)\right])^2}{\mathbb{E}\left[(g_\star(Z))^2\right]} \wedge \frac{\mathbb{E}\left[W \cdot g_\star(Z)\right]}{2}. \tag{55}$$

Next, we use that $g_\star$ satisfies (54), which implies that the second term in (55) is smaller, and hence,

$$\mathbb{E}\left[\log\left(1 + \lambda_\star^s f_\star^s(Z, W)\right)\right] \leq \frac{\mathbb{E}\left[W \cdot g_\star(Z)\right]}{2},$$

which concludes the proof of the second part of the theorem.

$$\square$$

**Corollary 1.** *Consider an arbitrary $g \in \mathcal{G}$ with nonnegative expected margin: $\mathbb{E}\left[W \cdot g(Z)\right] \geq 0$. Then the growth rate of the corresponding wealth process $(\mathcal{K}_t^s)_{t\geq0}$ satisfies:*

$$\liminf_{t\to\infty}\left(\tfrac{1}{t}\log\mathcal{K}_t^s\right) \overset{\text{a.s.}}{\geq} \tfrac{1}{4}\left(\sup_{s\in[0,1]}(1 - R_s(sg)) \wedge \mathbb{E}\left[W \cdot g(Z)\right]\right) \tag{18a}$$

$$\geq \tfrac{1}{4}\left(\mathbb{E}\left[W \cdot g(Z)\right]\right)^2, \tag{18b}$$

*and the optimal growth rate achieved by $\lambda_\star^s$ in (13) satisfies:*

$$\mathbb{E}\left[\log(1 + \lambda_\star^s f^s(Z, W))\right] \leq \left(\tfrac{4}{3} \cdot \sup_{s\in[0,1]}(1 - R_s(sg))\right) \wedge \left(\tfrac{1}{2} \cdot \mathbb{E}\left[W \cdot g(Z)\right]\right). \tag{19}$$

*Proof.* Following the same argument as that of the proof of Theorem 1, we can deduce that:

$$\liminf_{t\to\infty} \frac{\log \mathcal{K}_t^s}{t} \overset{\text{a.s.}}{\geq} \frac{1}{4}\left(\frac{(\mathbb{E}\left[W \cdot g(Z)\right])^2}{\mathbb{E}\left[g^2(Z)\right]} \wedge \mathbb{E}\left[W \cdot g(Z)\right]\right). \tag{56}$$

Hence, it suffices to argue that the lower bound (56) is equivalent to (18a). Without loss of generality, we can assume that $\mathbb{E}\left[W \cdot g(Z)\right] \geq 0$, and further, the two lower bounds are equal if $\mathbb{E}\left[W \cdot g(Z)\right] = 0$. Hence, we consider the case when $\mathbb{E}\left[W \cdot g(Z)\right] > 0$. First, let us consider the case when

$$\frac{\mathbb{E}\left[W \cdot g(Z)\right]}{\mathbb{E}\left[g^2(Z)\right]} < 1. \tag{57}$$

Using (51), we get that:

$$\sup_{s \in [0,1]} (1 - R_s(sg)) = \frac{(\mathbb{E}[W \cdot g(Z)])^2}{\mathbb{E}[g^2(Z)]},\tag{58}$$

and hence, two bounds coincide. For the upper bound (19), we use Lemma 6, and the upper bound (46) in particular. Note that the first term in (46) is less than the second term whenever

$$\frac{\mathbb{E}[W \cdot g(Z)]}{\mathbb{E}\left[(g(Z))^2\right]} \leq \frac{3}{8} < 1.$$

However, in this regime we also know that (58) holds, and hence the two bounds coincide. This completes the proof.

$\square$

**Theorem 2.** *The following claims hold for Seq-C-2ST (Algorithm 2):*

1. *If $H_0$ in (1a) is true, the test ever stops with probability at most $\alpha$: $\mathbb{P}_{H_0}(\tau < \infty) \leq \alpha$.*

2. *Suppose that $H_1$ in (1b) is true. Then:*

   (a) *Under Assumption 1, the test with the payoff (22a) is consistent: $\mathbb{P}_{H_1}(\tau < \infty) = 1$.*
   (b) *Under Assumption 2, the test with the payoff (22b) is consistent: $\mathbb{P}_{H_1}(\tau < \infty) = 1$.*

*Proof.* 1. We trivially have that the payoff functions (22a) and (22b) are bounded: $\forall t \geq 1$ and $\forall (z, w) \in \mathcal{Z} \times \{-1, 1\}$, it holds that $f_t^{\mathrm{m}}(z, w) \in [-1, 1]$ and $f_t^{\mathrm{s}}(z, w) \in [-1, 1]$. Further, under the null $H_0$ in (1a), it trivially holds that $\mathbb{E}_{H_0}[f_t^{\mathrm{m}}(Z_t, W_t) \mid \mathcal{F}_{t-1}] = \mathbb{E}_{H_0}[f_t^{\mathrm{s}}(Z_t, W_t) \mid \mathcal{F}_{t-1}] = 0$, where $\mathcal{F}_{t-1} = \sigma(\{(Z_i, W_i)\}_{i \leq t-1})$. Since ONS betting fractions $(\lambda_t^{\mathrm{ONS}})_{t \geq 1}$ are predictable, we conclude that the resulting wealth process is a nonnegative martingale. The assertion of the Theorem then follows directly from Ville's inequality (Proposition 2) when $a = 1/\alpha$.

2. Note that if ONS strategy for selecting betting fractions is deployed, then (49) implies that the tests will be consistent as long as

$$\liminf_{t \to \infty} \frac{1}{t} \sum_{i=1}^{t} f_i \overset{\mathrm{a.s.}}{>} 0,\tag{59}$$

where for $i \geq 1$, $f_i = f_i^{\mathrm{m}}(Z_i, W_i)$ and $f_i = f_i^{\mathrm{s}}(Z_i, W_i)$ for the payoffs based on the misclassification and the squared risks respectively.

(a) Recall that
$$f_i^{\mathrm{m}}(Z_i, W_i) = W_i \cdot \mathrm{sign}[g_i(Z_i)],$$
and Assumption 1 states that:

$$\limsup_{t \to \infty} \frac{1}{t} \sum_{i=1}^{t} \mathbb{1}\{W_i \cdot \mathrm{sign}[g_i(Z_i)] < 0\} \overset{\mathrm{a.s.}}{<} \frac{1}{2}.$$

Since $\mathbb{1}\{x < 0\} = (1 - \mathrm{sign}[x])/2$, we get that:

$$\limsup_{t \to \infty} \frac{1}{t} \sum_{i=1}^{t} \left(\frac{1}{2} - \frac{W_i \cdot \mathrm{sign}[g_i(Z_i)]}{2}\right) \overset{\mathrm{a.s.}}{<} \frac{1}{2},$$

which, after rearranging and multiplying by two, implies that:

$$\liminf_{t \to \infty} \frac{1}{t} \sum_{i=1}^{t} W_i \cdot \mathrm{sign}[g_i(Z_i)] \overset{\mathrm{a.s.}}{>} 0.$$

Hence, a sufficient condition for consistency (59) holds, and we conclude that the result is true.

(b) Recall that
$$f_i^{\mathrm{s}}(Z_i, W_i) = W_i \cdot g_i(Z_i),$$
and Assumption 2 states that:
$$\limsup_{t\to\infty} \frac{1}{t} \sum_{i=1}^{t} (g_i(Z_i) - W_i)^2 \overset{\mathrm{a.s}}{<} 1,$$
which is equivalent to
$$\limsup_{t\to\infty} \frac{1}{t} \sum_{i=1}^{t} \left(g_i^2(Z_i) - 2W_i \cdot g_i(Z_i)\right) \overset{\mathrm{a.s}}{<} 0.$$
It is easy to see that the above, in turn, implies that:
$$\liminf_{t\to\infty} \frac{1}{t} \sum_{i=1}^{t} W_i \cdot g_i(Z_i) \overset{\mathrm{a.s}}{>} 0.$$
Hence, a sufficient condition for consistency (59) holds, and we conclude that the result is true.

$\square$

## D.4  Proofs for Appendix A

**Theorem 3.** *The following claims hold for the oracle sequential regression-based IT based on* $\left(\mathcal{K}_t^{\mathrm{r},\star}\right)_{t\geq 0}$:

1. *Suppose that $H_0$ in (26a) is true. Then the test ever stops with probability at most $\alpha$:* $\mathbb{P}_{H_1}(\tau < \infty) \leq \alpha$.

2. *Suppose that $H_1$ in (26b) is true. Further, suppose that: $\mathbb{E}\left[W\ell(g_\star(X), Y)\right] > 0$. Then the test is consistent: $\mathbb{P}_{H_1}(\tau < \infty) = 1$.*

*Proof.*     1. We trivially have that the payoff function (27) is bounded: $\forall(x, y, w) \in \mathcal{X} \times \mathcal{Y} \times \{-1, 1\}$, it holds that $f_\star^{\mathrm{r}}(x, y, w) \in [-1, 1]$. Further, under the null $H_0$ in (26a), it trivially holds that $\mathbb{E}_{H_0}\left[f_\star^{\mathrm{r}}(X_t, Y_t, W_t) \mid \mathcal{F}_{t-1}\right] = 0$, where $\mathcal{F}_{t-1} = \sigma(\{(X_i, Y_i, W_i)\}_{i \leq t-1})$. Since ONS betting fractions $\left(\lambda_t^{\mathrm{ONS}}\right)_{t\geq 1}$ are predictable, we conclude that the resulting wealth process is a nonnegative martingale. The assertion of the Theorem then follows directly from Ville's inequality (Proposition 2) when $a = 1/\alpha$.

2. Note that if ONS strategy for selecting betting fractions is deployed, then (49) implies that the tests will be consistent as long as
$$\liminf_{t\to\infty} \frac{1}{t} \sum_{i=1}^{t} f_\star^{\mathrm{r}}(X_i, Y_i, W_i) \overset{\mathrm{a.s.}}{>} 0. \tag{60}$$

Note that:
$$\frac{1}{t} \sum_{i=1}^{t} f_\star^{\mathrm{r}}(X_i, Y_i, W_i) = \frac{1}{t} \sum_{i=1}^{t} \tanh\left(s_\star \cdot W_i \ell(g_\star(X_i), Y_i)\right) \overset{\mathrm{a.s.}}{\to} \mathbb{E}\left[\tanh\left(s_\star \cdot W\ell(g_\star(X), Y)\right)\right].$$

Note that for any $x \in \mathbb{R} : \tanh(x) \geq x - \frac{1}{3} \cdot \max\{x^3, 0\}$. Hence, for any $s > 0$, it holds that:

$$\mathbb{E}\left[\tanh\left(s \cdot W\ell(g_\star(X), Y)\right)\right] \geq s\mathbb{E}\left[W\ell(g_\star(X), Y)\right] - \frac{1}{3}\mathbb{E}\left[\max\left\{s^3 \cdot W(\ell(g_\star(X), Y))^3, 0\right\}\right]$$
$$= s\mathbb{E}\left[W\ell(g_\star(X), Y)\right] - \frac{s^3}{3}\mathbb{E}\left[(\ell(g_\star(X), Y))^3 \cdot \max\{W, 0\}\right]$$
$$= s\mathbb{E}\left[W\ell(g_\star(X), Y)\right] - \frac{s^3}{6}\mathbb{E}\left[(1 + W) \cdot (\ell(g_\star(X), Y))^3\right], \tag{61}$$

where we used that $\max\{W, 0\} = (W + 1)/2$ since $W \in \{-1, 1\}$. Maximizing the RHS of (61) over $s > 0$ yields $s_\star$ defined in (28a). Hence,

$$
\begin{aligned}
\mathbb{E}\left[\tanh\left(s_\star \cdot W\ell(g_\star(X), Y)\right)\right] &\geq s_\star\mathbb{E}\left[W\ell(g_\star(X), Y)\right] - \frac{s_\star^3}{6}\mathbb{E}\left[(1 + W) \cdot (\ell(g_\star(X), Y))^3\right] \\
&= s_\star\left(\mathbb{E}\left[W\ell(g_\star(X), Y)\right] - \frac{s_\star^2}{6}\mathbb{E}\left[(1 + W) \cdot (\ell(g_\star(X), Y))^3\right]\right) \\
&= s_\star\left(\mathbb{E}\left[W\ell(g_\star(X), Y)\right] - \frac{1}{3}\mathbb{E}\left[W\ell(g_\star(X), Y)\right]\right) \\
&= \frac{2s_\star}{3}\mathbb{E}\left[W\ell(g_\star(X), Y)\right] > 0.
\end{aligned}
$$

Hence, we conclude that the oracle regression-based IT is consistent since the sufficient condition (62) holds. $\qquad\square$

**Theorem 4.** *The following claims hold for the proxy sequential regression-based IT (Algorithm 3):*

1. *Suppose that $H_0$ in (26a) is true. Then the test ever stops with probability at most $\alpha$: $\mathbb{P}_{H_0}(\tau < \infty) \leq \alpha$.*

2. *Suppose that $H_1$ in (26b) is true. Further, suppose that Assumptions 3 and 4 are satisfied. Then the test is consistent: $\mathbb{P}_{H_1}(\tau < \infty) = 1$.*

*Proof.*     1. We trivially have that the payoff function (29) is bounded: $\forall(x, y, w) \in \mathcal{X} \times \mathcal{Y} \times \{-1, 1\}$, it holds that $f_t^r(x, y, w) \in [-1, 1]$. Further, under the null $H_0$ in (26a), it trivially holds that $\mathbb{E}_{H_0}[f_t^r(X_t, Y_t, W_t) \mid \mathcal{F}_{t-1}] = 0$, where $\mathcal{F}_{t-1} = \sigma(\{(X_i, Y_i, W_i)\}_{i \leq t-1})$. Since ONS betting fractions $(\lambda_t^{\mathrm{ONS}})_{t \geq 1}$ are predictable, we conclude that the resulting wealth process is a nonnegative martingale. The assertion of the Theorem then follows directly from Ville's inequality (Proposition 2) with $a = 1/\alpha$.

2. Note that if ONS strategy for selecting betting fractions is deployed, then (49) implies that the tests will be consistent as long as

$$
\liminf_{t \to \infty} \frac{1}{t}\sum_{i=1}^{t} f_t^r(X_i, Y_i, W_i) \overset{\mathrm{a.s.}}{>} 0. \tag{62}
$$

    (a) **Step 1.** Consider a predictable sequence of scaling factors $(s_t)_{t \geq 1}$, defined in (30a), and the corresponding sequences $(\mu_t)_{t \geq 1}$ and $(\nu_t)_{t \geq 1}$, defined in (30b) and (30c) respectively. For $t \geq 1$, let $\mathcal{F}_t := \sigma(\{(X_i, Y_i, W_i)\}_{i \leq t})$. Since the losses are bounded, we have that:

$$
\left(W_i \cdot \ell(g(X_i; \theta_i), Y_i) - \mathbb{E}\left[W_i \cdot \ell(g(X_i; \theta_i), Y_i) \mid \mathcal{F}_{i-1}\right]\right)_{i \geq 1},
$$

is a bounded martingale difference sequence (BMDS). By the Strong Law of Large Numbers for BMDS, it follows that:

$$
\frac{1}{t}\sum_{i=1}^{t}\left(W_i \cdot \ell(g(X_i; \theta_i), Y_i) - \mathbb{E}\left[W_i \cdot \ell(g(X_i; \theta_i), Y_i) \mid \mathcal{F}_{i-1}\right]\right) \overset{\mathrm{a.s.}}{\to} 0.
$$

Since $((X_t, Y_t, W_t))_{t \geq 1}$ is a sequence of i.i.d. observations, we can write

$$
\frac{1}{t}\sum_{i=1}^{t}\mathbb{E}\left[W_i \cdot \ell(g(X_i; \theta_i), Y_i) \mid \mathcal{F}_{i-1}\right] = \frac{1}{t}\sum_{i=1}^{t}\mathbb{E}\left[W \cdot \ell(g(X; \theta_i), Y) \mid \theta_i\right],
$$

where $(X, Y, W) \perp\!\!\!\perp (\theta_t)_{t \geq 1}, \theta_\star$. Using Assumption 3, we get that:

$$\left| \frac{1}{t} \sum_{i=1}^{t} \mathbb{E}\left[ W \cdot \ell(g(X; \theta_i), Y) \mid \theta_i \right] - \mathbb{E}\left[ W \cdot \ell(g(X; \theta_\star), Y) \mid \theta_\star \right] \right|$$

$$\leq \quad \frac{1}{t} \sum_{i=1}^{t} \sup_{\substack{x \in \mathcal{X} \\ y \in \mathcal{Y}}} |\ell(g(x; \theta_i), y) - \ell(g(x; \theta_\star), y)| \tag{63}$$

$$\leq \quad \frac{1}{t} \sum_{i=1}^{t} L_2 \sup_{x \in \mathcal{X}} |g(x; \theta_i) - g(x; \theta_\star)|$$

$$\leq \quad \frac{1}{t} \sum_{i=1}^{t} L_2 \cdot L_1 \cdot \|\theta_i - \theta_\star\| \overset{\text{a.s.}}{\to} 0,$$

since $\|\theta_i - \theta_\star\| \overset{\text{a.s.}}{\to} 0$ by Assumption 4. In particular, this implies that $\mu_t \overset{\text{a.s.}}{\to} \mathbb{E}\left[ W\ell(g(X; \theta_\star), Y) \mid \theta_\star \right]$. Similar argument can be used to show that $\nu_t \overset{\text{a.s.}}{\to} \mathbb{E}\left[ (1 + W) \cdot (\ell(g(X; \theta_\star), Y))^3 \mid \theta_\star \right]$, and hence,

$$s_t \overset{\text{a.s.}}{\to} \sqrt{\frac{2\mathbb{E}\left[ W\ell(g(X; \theta_\star), Y) \mid \theta_\star \right]}{\mathbb{E}\left[ (1 + W) \cdot (\ell(g(X; \theta_\star), Y))^3 \mid \theta_\star \right]}} =: s_\star. \tag{64}$$

Note that $s_\star$ is a random variable which is positive (almost surely) by Assumption 4.

(b) **Step 2.** Recall that for any $x \in \mathbb{R} : \tanh(x) \geq x - \frac{1}{3} \cdot \max\{x^3, 0\}$ and that $\max\{W, 0\} = (W + 1)/2$ since $W \in \{-1, 1\}$. We have:

$$\frac{1}{t} \sum_{i=1}^{t} f_i^{\text{r}}(X_i, Y_i, W_i)$$

$$= \quad \frac{1}{t} \sum_{i=1}^{t} \tanh\left( s_i \cdot W_i \ell(g(X_i; \theta_i), Y_i) \right)$$

$$\geq \quad \frac{1}{t} \sum_{i=1}^{t} \left( s_i \cdot W_i \cdot \ell(g(X_i; \theta_i), Y_i) - \frac{s_i^3}{6} \cdot (1 + W_i) \cdot (\ell(g(X_i; \theta_i), Y_i))^3 \right).$$

Note that $\theta_i$ and $s_i$ are $\mathcal{F}_{i-1}$-measurable (see Step 1 for the definition of $\mathcal{F}_{i-1}$). Under a minor technical assumption that $(s_t)_{t \geq 1}$ is a sequence of bounded scaling factors (the lower bound is trivially zero and the upper bound also holds if $\nu_t$ are bounded away from zero almost surely which is reasonable given the definition of $\nu_t$), we can use analogous argument regarding a BMDS in Step 1 to deduce that:

$$\liminf_{t \to \infty} \frac{1}{t} \sum_{i=1}^{t} f_i^{\text{r}}(X_i, Y_i, W_i)$$

$$\geq \quad \liminf_{t \to \infty} \frac{1}{t} \sum_{i=1}^{t} \left( s_i \cdot \mathbb{E}\left[ W \cdot \ell(g(X; \theta_i), Y) \mid \theta_i \right] - \frac{s_i^3}{6} \mathbb{E}\left[ (1 + W) \cdot (\ell(g(X; \theta_i), Y))^3 \mid \theta_i \right] \right). \tag{65}$$

Using argument analogous to (63), we can show that:

$$\frac{1}{t} \sum_{i=1}^{t} \mathbb{E}\left[ (1 + W) \cdot (\ell(g(X; \theta_i), Y))^3 \mid \theta_i \right] \overset{\text{a.s.}}{\to} \mathbb{E}\left[ (1 + W) \cdot (\ell(g(X; \theta_\star), Y))^3 \mid \theta_\star \right]. \tag{66}$$

Combining (63), (64) and (66), we deduce that

$$\frac{1}{t} \sum_{i=1}^{t} \left( s_i \cdot \mathbb{E}\left[ W \cdot \ell(g(X; \theta_i), Y) \mid \theta_i \right] - \frac{s_i^3}{6} \mathbb{E}\left[ (1 + W) \cdot (\ell(g(X; \theta_i), Y))^3 \mid \theta_i \right] \right)$$

$$\overset{\text{a.s.}}{\to} \quad s_\star \cdot \mathbb{E}\left[ W \cdot \ell(g(X; \theta_\star), Y) \mid \theta_\star \right] - \frac{s_\star^3}{6} \cdot \mathbb{E}\left[ (1 + W) \cdot (\ell(g(X; \theta_\star), Y))^3 \mid \theta_\star \right]$$

$$= \frac{2s_\star}{3} \cdot \mathbb{E}\left[W \cdot \ell(g(X;\theta_\star), Y) \mid \theta_\star\right].$$

Hence, from (65) it follows that:

$$\liminf_{t\to\infty} \frac{1}{t} \sum_{i=1}^{t} f_i^{\mathrm{r}}(X_i, Y_i, W_i) \geq \frac{2s_\star}{3} \cdot \mathbb{E}\left[W \cdot \ell(g(X;\theta_\star), Y) \mid \theta_\star\right],$$

where the RHS is a random variable which is positive almost surely. Hence, a sufficient condition for consistency (62) holds which concludes the proof.

$\square$

### D.5 Proofs for Appendix B

**Two-Sample Testing with Unbalanced Classes.** Note that $(g(z) = 2\eta(z) - 1)$:

$$(1 - \lambda_t) \cdot 1 + \lambda_t \cdot \frac{(\eta(Z_t))^{\mathbb{1}\{W_t=1\}} (1 - \eta(Z_t))^{1-\mathbb{1}\{W_t=1\}}}{(\pi)^{\mathbb{1}\{W_t=1\}} (1 - \pi)^{1-\mathbb{1}\{W_t=1\}}}$$

$$= (1 - \lambda_t) \cdot 1 + \lambda_t \cdot \frac{\left(\frac{1+g(Z_t)}{2}\right)^{\mathbb{1}\{W_t=1\}} \left(\frac{1-g(Z_t)}{2}\right)^{1-\mathbb{1}\{W_t=1\}}}{(\pi)^{\mathbb{1}\{W_t=1\}} (1 - \pi)^{1-\mathbb{1}\{W_t=1\}}}$$

$$= (1 - \lambda_t) \cdot 1 + \frac{\lambda_t}{2} \cdot \frac{(1 + g(Z_t))^{\mathbb{1}\{W_t=1\}} (1 - g(Z_t))^{1-\mathbb{1}\{W_t=1\}}}{(\pi)^{\mathbb{1}\{W_t=1\}} (1 - \pi)^{1-\mathbb{1}\{W_t=1\}}}$$

$$= (1 - \lambda_t) \cdot 1 + \frac{\lambda_t}{2} \cdot \frac{1 + W_t g(Z_t)}{(\pi)^{\mathbb{1}\{W_t=1\}} (1 - \pi)^{1-\mathbb{1}\{W_t=1\}}}$$

$$= (1 - \lambda_t) \cdot 1 + \frac{\lambda_t}{2} \cdot \frac{2}{1 + W_t(2\pi - 1)} \cdot (1 + W_t g(Z_t))$$

$$= (1 - \lambda_t) \cdot 1 + \frac{\lambda_t}{1 + W_t(2\pi - 1)} \cdot (1 + W_t g(Z_t))$$

$$= 1 + \lambda_t \cdot \frac{W_t(g(Z_t) - (2\pi - 1))}{1 + W_t(2\pi - 1)}.$$

**Payoff for the Case of Unbalanced Classes (known $\pi$).** To see that the payoff function (37) is lower bounded by negative one, note that:

$$f_t^{\mathrm{u}}(z, 1) = \frac{g_t(z) - (2\pi - 1)}{2\pi} \geq \frac{-1 - (2\pi - 1)}{2\pi} = -1,$$

$$f_t^{\mathrm{u}}(z, -1) = \frac{-g_t(z) + (2\pi - 1)}{2(1 - \pi)} \geq \frac{-1 + (2\pi - 1)}{2(1 - \pi)} = -1.$$

To see that such payoff is fair, note that:

$$\mathbb{E}_{H_0}\left[f_t^{\mathrm{u}}(Z_t, W_t) \mid \mathcal{F}_{t-1}\right] = \mathbb{E}_P\left[\pi \cdot \frac{g_t(Z_t) - (2\pi - 1)}{2\pi}\right] - \mathbb{E}_Q\left[(1 - \pi) \cdot \frac{g_t(Z_t) - (2\pi - 1)}{2(1 - \pi)} \mid \mathcal{F}_{t-1}\right] = 0,$$

where $\mathcal{F}_{t-1} = \sigma\left(\{(Z_i, W_i)\}_{i \leq t-1}\right)$.

**Theorem 5.** *Suppose that $H_0$ in (35a) is true. Then $(\mathcal{K}_t^{\mathrm{u}})_{t\geq 0}$ is a nonnegative supermartingale adapted to $(\mathcal{F}_t)_{t\geq 0}$. Hence, the sequential 2ST based on $(\mathcal{K}_t^{\mathrm{u}})_{t\geq 0}$ satisfies: $\mathbb{P}_{H_0}(\tau < \infty) \leq \alpha$.*

*Proof.* First, we show that $(\mathcal{K}_t^{\mathrm{u}})_{t\geq 0}$ is a nonnegative supermartingale. For any $t \geq 1$, the wealth $\mathcal{K}_{t-1}$ is multiplied at round $t$ by

$$1 + \lambda_t f_t^{\mathrm{u}}\left(\{(Z_{b(t-1)+i}, W_{b(t-1)+i})\}_{i\in\{1,\ldots,b\}}\right) = (1 - \lambda_t) \cdot 1 + \lambda_t \cdot \frac{\prod_{i=b(t-1)+1}^{bt} (1 + W_i g_t(Z_i))}{\prod_{i=1}^{b} (1 + W_i(2\hat{\pi}_t - 1))}.$$

Since $\lambda_t \in [0, 0.5]$, we conclude that the process $(\mathcal{K}_t^{\mathrm{u}})_{t \geq 0}$ is nonnegative. Next, note that since $\hat{\pi}_t$ is the MLE of $\pi$ computed from a $t$-th minibatch, it follows that:

$$1 + \lambda_t f_t^{\mathrm{u}} \left( \left\{ (Z_{b(t-1)+i}, W_{b(t-1)+i}) \right\}_{i \in \{1, \ldots, b\}} \right) \leq (1 - \lambda_t) \cdot 1 + \lambda_t \cdot \frac{\prod_{i=b(t-1)+1}^{bt} (1 + W_i g_t(Z_i))}{\prod_{i=b(t-1)+1}^{bt} (1 + W_i(2\pi - 1))}$$

$$= (1 - \lambda_t) \cdot 1 + \lambda_t \cdot \prod_{i=b(t-1)+1}^{bt} \left( \frac{1 + W_i g_t(Z_i)}{1 + W_i(2\pi - 1)} \right).$$

Recall that $\mathcal{F}_{t-1} = \sigma \left( \{ Z_i, W_i \}_{i \leq b(t-1)} \right)$. It suffices to show that if $H_0$ is true, then

$$\mathbb{E}_{H_0} \left[ \prod_{i=b(t-1)+1}^{bt} \left( \frac{1 + W_i g_t(Z_i)}{1 + W_i(2\pi - 1)} \right) \mid \mathcal{F}_{t-1} \right] = 1.$$

Note that the individual terms in the above product are independent conditional on $\mathcal{F}_{t-1}$. Hence,

$$\mathbb{E}_{H_0} \left[ \prod_{i=b(t-1)+1}^{bt} \left( \frac{1 + W_i g_t(Z_i)}{1 + W_i(2\pi - 1)} \right) \mid \mathcal{F}_{t-1} \right] = \prod_{i=b(t-1)+1}^{bt} \mathbb{E}_{H_0} \left[ \frac{1 + W_i g_t(Z_i)}{1 + W_i(2\pi - 1)} \mid \mathcal{F}_{t-1} \right].$$

For any $i \in \{b(t-1)+1, \ldots, bt\}$, it holds that:

$$\mathbb{E}_{H_0} \left[ \frac{1 + W_i g_t(Z_i)}{1 + W_i(2\pi - 1)} \mid \mathcal{F}_{t-1} \right] = \mathbb{E}_{H_0} \left[ \pi \cdot \frac{1 + g_t(Z_i)}{1 + (2\pi - 1)} + (1 - \pi) \cdot \frac{1 - g_t(Z_i)}{1 - (2\pi - 1)} \mid \mathcal{F}_{t-1} \right]$$

$$= \mathbb{E}_{H_0} \left[ \frac{1 + g_t(Z_i)}{2} + \frac{1 - g_t(Z_i)}{2} \mid \mathcal{F}_{t-1} \right]$$

$$= 1.$$

Hence, we conclude that $(\mathcal{K}_t^{\mathrm{u}})_{t \geq 0}$ is a nonnegative supermartingale adapted to $(\mathcal{F}_t)_{t \geq 0}$. The time-uniform type I error control of the resulting test then follows from Ville's inequality (Proposition 2). $\square$

## E    Additional Experiments and Details

### E.1    Modeling Details

**CNN Architecture and Training.**    We use CNN with 4 convolutional layers (kernel size is taken to be $3 \times 3$) and 16, 32, 32, 64 filters respectively. Further, each convolutional layer is followed by max-pooling layer ($2 \times 2$). After flattening, those layers are followed by 1 fully connected layer with 128 neurons. Dropout ($p = 0.5$) and early stopping (with patience equal to ten epochs and 20% of data used in the validation set) is used for regularization. ReLU activation functions are used in each layer. Adam optimizer is used for training the network. We start training after processing twenty observations, and update the model parameters after processing every next ten observations. Maximum number of epochs is set to 25 for each training iteration. The batch size is set to 32.

**Single-stream Sequential Kernelized 2ST.**    The construction of this test is the extension of 2ST of Shekhar and Ramdas [2023] to the case when at each round an observation only from a single distribution ($P$ or $Q$) is revealed. Let $\mathcal{G}$ denote an RKHS with positive-definite kernel $k$ and canonical feature map $\varphi(\cdot)$ defined on $\mathcal{Z}$. Recall that instances from $P$ as labeled as $+1$ and instances from $Q$ are labeled as $-1$ (characterized by $W$). The mean embeddings of $P$ and $Q$ are then defined as

$$\hat{\mu}_P^{(t)} = \frac{1}{N_+(t)} \sum_{i=1}^{t} \varphi(Z_i) \cdot \mathbb{1} \{ W_i = +1 \},$$

$$\hat{\mu}_Q^{(t)} = \frac{1}{N_-(t)} \sum_{i=1}^{t} \varphi(Z_i) \cdot \mathbb{1} \{ W_i = -1 \},$$

where $N_+(t) = |i \le t : W_i = +1|$ and $N_-(t) = |i \le t : W_i = -1|$. The corresponding payoff function is

$$f_t^{\mathrm{k}}(Z_{t+1}, W_{t+1}) = W_{t+1} \cdot \hat{g}_t(Z_{t+1}),$$

$$\text{where} \quad \hat{g}_t = \frac{\hat{\mu}_P^{(t)} - \hat{\mu}_Q^{(t)}}{\left\| \hat{\mu}_P^{(t)} - \hat{\mu}_Q^{(t)} \right\|_{\mathcal{G}}}.$$

To make the test computationally efficient, it is critical to update the normalization constant efficiently. Suppose that at round $t + 1$, an instance from $P$ is observed. In this case, $\hat{\mu}_Q^{(t+1)} = \hat{\mu}_Q^{(t)}$. Note that:

$$\hat{\mu}_P^{(t+1)} = \frac{1}{N_+(t+1)} \sum_{i=1}^{t+1} \varphi(Z_i) \cdot \mathbb{1}\{W_i = +1\}$$

$$= \frac{1}{N_+(t)+1} \sum_{i=1}^{t+1} \varphi(Z_i) \cdot \mathbb{1}\{W_i = +1\}$$

$$= \frac{1}{N_+(t)+1} \varphi(Z_{t+1}) + \frac{1}{N_+(t)+1} \sum_{i=1}^{t} \varphi(Z_i) \cdot \mathbb{1}\{W_i = +1\}$$

$$= \frac{1}{N_+(t)+1} \varphi(Z_{t+1}) + \frac{N_+(t)}{N_+(t)+1} \hat{\mu}_P^{(t)}.$$

Hence, we have:

$$\left\| \hat{\mu}_P^{(t+1)} - \hat{\mu}_Q^{(t+1)} \right\|_{\mathcal{G}}^2 = \left\| \hat{\mu}_P^{(t+1)} - \hat{\mu}_Q^{(t)} \right\|_{\mathcal{G}}^2$$

$$= \left\| \hat{\mu}_P^{(t+1)} \right\|_{\mathcal{G}}^2 - 2 \left\langle \hat{\mu}_P^{(t+1)}, \hat{\mu}_Q^{(t)} \right\rangle_{\mathcal{G}} + \left\| \hat{\mu}_Q^{(t)} \right\|_{\mathcal{G}}^2.$$

In particular,

$$\left\langle \hat{\mu}_P^{(t+1)}, \hat{\mu}_Q^{(t)} \right\rangle_{\mathcal{G}} = \left\langle \frac{1}{N_+(t)+1} \varphi(Z_{t+1}) + \frac{N_+(t)}{N_+(t)+1} \hat{\mu}_P^{(t)}, \hat{\mu}_Q^{(t)} \right\rangle_{\mathcal{G}}$$

$$= \frac{1}{N_+(t)+1} \left\langle \varphi(Z_{t+1}), \hat{\mu}_Q^{(t)} \right\rangle_{\mathcal{G}} + \frac{N_+(t)}{N_+(t)+1} \left\langle \hat{\mu}_P^{(t)}, \hat{\mu}_Q^{(t)} \right\rangle_{\mathcal{G}}.$$

Note that:

$$\left\langle \varphi(Z_{t+1}), \hat{\mu}_Q^{(t)} \right\rangle_{\mathcal{G}} = \frac{1}{N_-(t)} \sum_{i=1}^{t} k(Z_{t+1}, Z_i) \cdot \mathbb{1}\{W_i = -1\}.$$

Next, we assume for simplicity that $k(x, x) = 1, \forall x$ which holds for RBF kernel. Observe that:

$$\left\| \hat{\mu}_P^{(t+1)} \right\|_{\mathcal{G}}^2 = \left\langle \hat{\mu}_P^{(t+1)}, \hat{\mu}_P^{(t+1)} \right\rangle_{\mathcal{G}}$$

$$= \frac{1}{(N_+(t)+1)^2} + \frac{2N_+(t)}{(N_+(t)+1)^2} \left\langle \varphi(Z_{t+1}), \hat{\mu}_P^{(t)} \right\rangle_{\mathcal{G}} + \frac{(N_+(t))^2}{(N_+(t)+1)^2} \left\| \hat{\mu}_P^{(t)} \right\|_{\mathcal{G}}^2.$$

By caching intermediate results, we can compute the normalization constant using linear in $t$ number of kernel evaluations. We start betting once at least one instance is observed from both $P$ and $Q$. For simulations, we use RBF kernel and the median heuristic with first 20 instances to compute the kernel hyperparameter.

**MLP Training Scheme**   We begin training after processing twenty datapoints from $P_{XY}$ which gives a training dataset with 40 datapoints (due to randomization). When updating a model, we use previous parameters as initialization. We use the following update scheme: we start after next $n_0 = 10$ datapoints from $P_{XY}$ are observed. Once $n_0$ becomes less than 1% of the size of the existing training dataset, we increase it by ten, that is, $n_t = n_{t-1} + 10$. When we fit the model, we set the maximum number of epochs to be 25 and use early stopping with patience of 3 epochs.

**Kernel Hyperparameters for Synthetic Experiments.** For SKIT, we use RBF kernels:

$$k(x, x') = \exp\left(-\lambda_X \|x - x'\|_2^2\right), \quad l(y, y') = \exp\left(-\lambda_Y \|y - y'\|_2^2\right).$$

For simulations on synthetic data, we take kernel hyperparameters to be inversely proportional to the second moment of the underlying variables (the median heuristic yields similar results):

$$\lambda_X = \frac{1}{2\mathbb{E}\left[\|X - X'\|_2^2\right]}, \quad \lambda_Y = \frac{1}{2\mathbb{E}\left[\|Y - Y'\|_2^2\right]}.$$

1. *Spherical model.* By symmetry, we have: $P_X = P_Y$, and hence we take $\lambda_X = \lambda_Y$. We have

$$\mathbb{E}\left[(X - X')^2\right] = 2\mathbb{E}\left[X^2\right] = \frac{2}{d}.$$

2. *HTDD model.* By symmetry, we have: $P_X = P_Y$, and hence we take $\lambda_X = \lambda_Y$. We have

$$\mathbb{E}\left[(X - X')^2\right] = 2\mathbb{E}\left[X^2\right] = \frac{2\pi^2}{3}.$$

3. *Sparse signal model.* We have

$$\mathbb{E}\left[\|X - X'\|_2^2\right] = 2\mathbb{E}\left[\|X\|_2^2\right] = 4d,$$

$$\mathbb{E}\left[\|Y - Y'\|_2^2\right] = 2\mathbb{E}\left[\|Y\|_2^2\right] = 2\mathrm{tr}(B_s B_s^\top + I_d) = 2(d + \sum_{i=1}^{d} \beta_i^2).$$

4. *Gaussian model.* We have

$$\mathbb{E}\left[(X - X')^2\right] = 2\mathbb{E}\left[X^2\right] = 2,$$
$$\mathbb{E}\left[(Y - Y')^2\right] = 2\mathbb{E}\left[Y^2\right] = 2(1 + \beta^2).$$

**Ridge Regression.** We use ridge regression as an underlying predictive model: $\hat{g}_t(x) = \beta_0^{(t)} + x\beta_1^{(t)}$, where the coefficients are obtained by solving:

$$(\beta_0^{(t)}, \beta_1^{(t)}) = \arg\min_{\beta_0, \beta_1} \sum_{i=1}^{2(t-1)} (Y_i - X_i\beta_1 - \beta_0)^2 + \lambda\beta_1^2.$$

Let $\Gamma = \mathrm{diag}(0, 1)$. Let $\mathbf{X}_{t-1} \in \mathbb{R}^{2(t-1)\times 2}$ be such that $(\mathbf{X}_{t-1})_i = (1, X_i)$, $i \in [1, 2(t-1)]$. Finally, let $\mathbf{Y}_{t-1}$ be a vector of responses: $(\mathbf{Y}_{t-1})_i = Y_i$, $i \in [1, 2(t-1)]$. Then:

$$\beta^{(t)} = \arg\min_{\beta} \|\mathbf{Y}_{t-1} - \mathbf{X}_{t-1}\beta\|^2 + \lambda\beta^\top\Gamma\beta = \left(\mathbf{X}_{t-1}^\top\mathbf{X}_{t-1} + \lambda\Gamma\right)^{-1}\left(\mathbf{X}_{t-1}^\top\mathbf{Y}_{t-1}\right).$$

## E.2 Additional Experiments for Seq-C-IT

In Figure 7, we present average stopping times for ITs under the synthetic settings from Section 3. We confirm that all tests adapt to the complexity of a problem at hand, stopping earlier on easy tasks and later on harder ones. We also consider two additional synthetic examples where Seq-C-IT outperforms a kernelized approach:

1. *Sparse signal model.* Let $(X_t)_{t\geq 1}$ and $(\varepsilon_t)_{t\geq 1}$ be two independent sequences of standard Gaussian random vectors in $\mathbb{R}^d$: $X_t, \varepsilon_t \overset{\mathrm{iid}}{\sim} \mathcal{N}(0, \mathbf{I}_d)$, $t \geq 1$. We take

$$(X_t, Y_t) = (X_t, B_s X_t + \varepsilon_t),$$

where $B_s = \mathrm{diag}(\beta_1, \ldots, \beta_d)$ and only $s = 5$ of $\{\beta_i\}_{i=1}^{d}$ are nonzero being sampled from $\mathrm{Unif}([-0.5, 0.5])$. We consider $d \in \{5, \ldots, 50\}$.

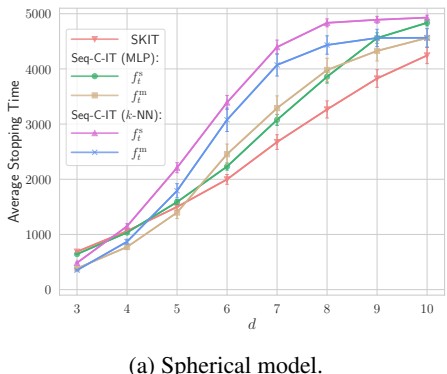 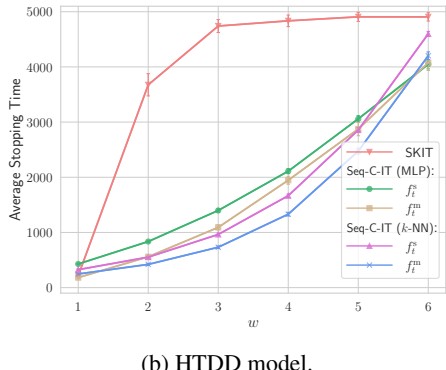

|     |     |
| --- | --- |
| (a) Spherical model. | (b) HTDD model. |

Figure 7: Stopping times of ITs on synthetic data from Section 3. Subplot (a) shows that SKIT is only marginally better than Seq-C-IT (MLP) due to slightly better sample efficiency under the spherical model (no localized dependence). Under the structured HTDD model, SKIT is inferior to Seq-C-ITs.

2. *Nested circles model.* Let $(L_t)_{t \geq 1}$, $(\Theta_t)_{t \geq 1}$, $(\varepsilon_t^{(1)})_{t \geq 1}$, $(\varepsilon_t^{(2)})_{t \geq 1}$ denote sequences of random variables where $L \overset{\text{iid}}{\sim} \text{Unif}(1, \ldots, l)$ for some prespecified $l \in \mathbb{N}$, $\Theta_t \overset{\text{iid}}{\sim} \text{Unif}([0, 2\pi])$, and $\varepsilon_t^{(1)}, \varepsilon_t^{(2)} \overset{\text{iid}}{\sim} \mathcal{N}(0, (1/4)^2)$. For $t \geq 1$, we take

$$(X_t, Y_t) = (L_t \cos(\Theta_t) + \varepsilon_t^{(1)}, L_t \sin(\Theta_t) + \varepsilon_t^{(2)}). \tag{67}$$

We consider $l \in \{1, \ldots, 10\}$.

In Figure 8, we show that Seq-C-ITs significantly outperform SKIT under these models. We note that the degrading performance of kernel-based tests under the nested circles model (67) has been also observed in earlier works [Berrett and Samworth, 2019, Podkopaev et al., 2023].

### E.3 Sequential vs Batch 2STs

The role of sequential tests is to complement existing batch tests and not replace those. Generally, if one is ready to commit to a particular sample size, it is better to refer to batch tests. To understand the loss of power, we conducted an experiment where we take $P = \mathcal{N}(0, 1)$, $Q = \mathcal{N}(\delta, 1)$ and use $k$-NN predictor as an underlying predictor. As we point out in Remark 1, if an analyst stops the experiment early, i.e., before the wealth exceeds $1/\alpha$, there is one modification that allows using non-exhausted type-I error budget to strictly improve power [Ramdas and Manole, 2023]: one may use a different threshold for rejecting the null, namely $U/\alpha$, where $U$ is an independently drawn (stochastically larger than) uniform random variable on $[0, 1]$.

We set the sample size to 1500 and compared three tests: batch (with half of the data used for training and half for inference), Seq-C-2ST (our sequential test), and Seq-C-2ST + Randomization (as per Remark 1). In Figure 9, we observe that the batch test has only slightly higher power than our sequential one when the sample size is specified beforehand. Yet in cases where the power is less than one using a sequential test allows collecting more data to improve it, but with the batch test, nothing can be done since the type-1 error budget is fully exhausted.

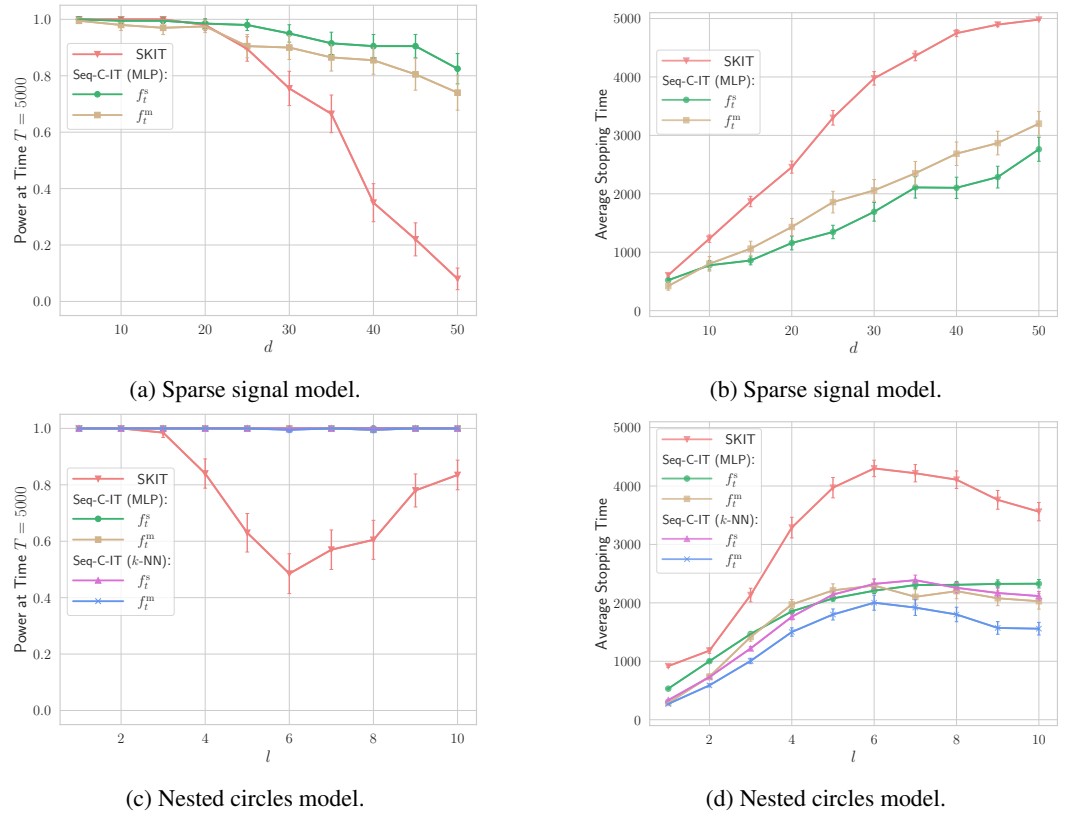

(a) Sparse signal model.

(b) Sparse signal model.

(c) Nested circles model.

(d) Nested circles model.

Figure 8: Rejection rates (left column) and average stopping times (right column) of sequential ITs for synthetic datasets from Appendix E.2. In both cases, SKIT is inferior to Seq-C-ITs.

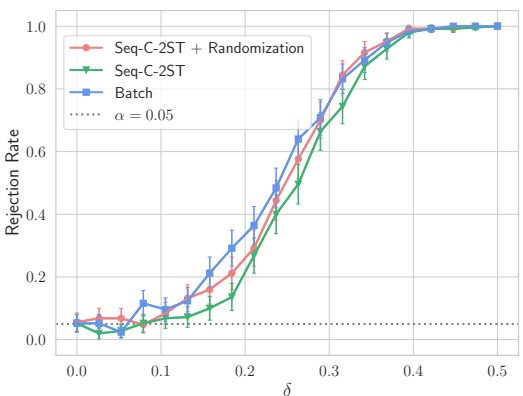

Figure 9: Comparison between the batch and sequential 2STs on Gaussian data when the sample size is set beforehand to 1500. First, invoking randomization improves the power of our sequential test across all settings. Second, batch 2ST has only slightly higher power across many settings. Note that the power of Seq-C-2ST can be easily improved after collecting a bit more data, yet this is not allowed for the batch test.

