# OpenReview forum: "Sequential Predictive Two-Sample and Independence Testing"
_NeurIPS.cc/2023/Conference — NeurIPS 2023 poster_

### Official Review · Reviewer_qWK6 · 2023-06-16

**Soundness:** 4 excellent
**Presentation:** 3 good
**Contribution:** 3 good
**Rating:** 8
**Confidence:** 4

**Summary:**

[Update: After reading the other reviews and the rebuttal, I have increased my score from 7 to 8]

The paper provides an algorithm for sequential testing of the homogeneity (two-sample) and independence hypothesis.
The core idea is that one learns a classifier to distinguish $P$ from $Q$ (in case of two-sample testing) or $P_{XY}$ from $P_X \times P_Y$ (in case of Independence testing) from the previously seen data. This learned algorithm is used to place a bet on a new observation (how likely it is that it is from one class or the other). If it matches one updates the betting budget (starting from 1). There is also a parameter that trades-off how much of the existing budget is used at a step. If the betting budget ever exceeds $1/\alpha$, with $\alpha$ the significance level, then one rejects the test.

The authors show that the test controls type-I error and is consistent under the alternative if a non-trivial classifier can be learned. Furthermore they provide a strategy to select betting fractions which improves performance, especially if the classifier during the first iterations has poor quality.



**Strengths:**

- The paper nicely combines recent work on sequential testing with the power of general classification algorithms. It does not require any specific learning algorithms but can use any existing learning framework.
- The theory is presented at an adequate level.
- The experiments nicely underline the theoretical statements of the work and show good empirical performance.


**Weaknesses:**

- If I understand correctly, the main innovation over Pandeva et al [2022] is that the present work uses adaptive betting fractions? Is this a correct interpretation? I struggle a bit to clearly see the parallels and differences to Pandeva et al. They formulate their test in terms of E-values. Can your tests also be rephrased in terms of E-values to make a comparison simpler?



**Minor**:
- It took me a bit to understand the notation in some places:
  - l. 75. What does $\sigma$ stand for?
  - After l. 104. Say that probability and expectations are wrt 1/2 (P+Q) (at least that's what I figured)
  - Introduce $\vee \wedge$ notation for min/max
  - l. 127 what is the expectation taken over?

**Questions:**

- Please see my remark above regarding Pandeva et al.

Below are curiosities that could further improve the paper, but are not absolutely critical.
- In Batch C2ST, where the sample size is fixed upfront, usually half the data is used for learning/testing. However, I am not aware that this is a principled choice. What if you used your approach even in batch testing to circumvent the need of selecting the splitting ratio. Can we get better power? I think it would be nice to have an experiment with exactly the same algorithm  / architecture and compare the two approaches.
- You prove that the type-I error is controlled even if the test is run indefinitely. I am curious: Does the test eventually exhaust the significance level? Do you have any theoretical or empirical insights?
- you apply your test to every new datapoint. Panderas et al only update after a small batch of new data is presented. Have you considered this? It might be computationally a bit more convenient, and I am curious how it would affect test power.

**Limitations:**

I do not see much limitations.

---

> ### Author Rebuttal · Authors · 2023-08-08
>
> We thank the reviewer for constructive feedback on the paper. Below, we address in detail issues that have been raised in the review.
>
> > The main innovation over Pandeva et al [2022] is that the present work uses adaptive betting fractions?...hard to clearly see the parallels and differences to Pandeva et al. They formulate their test in terms of E-values. Can your tests also be rephrased in terms of E-values to make the comparison simpler?
>
> Our works are indeed closely related as we pointed out in the paper. However, our focus is on the sequential setting, whereas Pandeva et al. focus on the batch case. Our construction results in a nonnegative martingale starting at 1 under the null hypothesis (test martingale). Test martingales are essentially a dynamic version of (conditional) e-values (see [1] for more context). The critical difference is indeed in using adaptive betting fractions which results in (a) substantial empirical improvements in terms of power as we illustrate in Example 1 in the paper, (b) much milder conditions for consistency (Pandeva et al. do not actually even discuss consistency, but the conditions should essentially the same as for the test of Lhéritier and Cazals, whereas our test is consistent as long as the squared errors (or misclassification error for the second test) are better on average than that of a naive predictor).
>
> [1] ``Game-Theoretic Statistics and Safe Anytime-Valid Inference'', Ramdas et al., 2023
>
> > Line 75: what does $\sigma$ stand for?
>
> $\mathcal{F}_t = \sigma(Z_1,\dots,Z_t)$ stands for the sigma-field generated by $Z_1,\dots,Z_t$, with $\mathcal{F}_0$ standing for the trivial sigma-field.
>
> > After line 104. Say that probability and expectations are wrt 1/2 (P+Q) (at least that's what I figured)
>
> Section 2 indeed corresponds to the setting of Definition 1. We have clarified that point in the revision.
>
> > Introduce $\wedge, \vee$ notation for min/max
>
> Done, thanks.
>
> > Line 127 what is the expectation taken over?
>
> Section 2 corresponds to the setting of Definition 1, i.e., $(Z_1, W_1),(Z_2, W_2),\dots$ (and $(Z,W)$) are i.i.d. draws from $\frac{1}{2}(P+Q)$, and those are the only sources of randomness over which expectations are taken over. We have clarified that point in the revision.
>
> > You prove that the type-1 error is controlled even if the test is run indefinitely. I am curious: Does the test eventually exhaust the significance level? Do you have any theoretical or empirical insights?
>
> Yes, the test "essentially" exhausts the significance level (Ville's inequality holds with equality for continuous time martingales if the limiting cumulative variance is infinite, which it will be if we never stop betting; in discrete time, there is a slight looseness due to ``overshoot" but it is a second order effect).
>
> In practice, it is indeed the case that the tests can not be run indefinitely. As we point out in Remark 1, if an analyst stops the experiment early, i.e., before the wealth exceeds $1/\alpha$, there is one modification that allows using non-exhausted type-I error budget to strictly improve power: one may use a different threshold for rejecting the null, namely $U/\alpha$, where $U$ is an independently drawn (stochastically larger than) uniform random variable on [0, 1]; see [2] for more details.
>
> [2] ``Randomized and exchangeable improvements of Markov’s, Chebyshev’s and Chernoff’s inequalities'', Ramdas and Manole, 2023.
>
> > In Batch C2ST, where the sample size is fixed upfront, usually half the data is used for learning/testing. However, I am not aware that this is a principled choice. What if you used your approach even in batch testing to circumvent the need of selecting the splitting ratio. Can we get better power? I think it would be nice to have an experiment with exactly the same algorithm/architecture and compare the two approaches.
>
> The role of sequential tests is to complement existing batch tests and not replace those. Generally, if one is ready to commit to a particular sample size, it is better to refer to batch tests. To understand the loss of power, we conducted an experiment, and the results are added to the allowed separate PDF. We take $P=\mathcal{N}(0,1)$, $Q=\mathcal{N}(\delta,1)$, and use $k$-NN predictor. We set the sample size to 1500 and compared three tests: batch (with half of the data used for training and half for inference), Seq-C-2ST (our sequential test), and Seq-C-2ST + Randomization (as per Remark 1). The batch test has (slightly) higher power than our sequential one if the sample size is specified beforehand. Yet in cases where the power is less than one using a sequential test allows collecting more data to improve it, but with the batch test, nothing can be done since the type-1 error budget is fully utilized. We have added this experiment to the Appendix and included the figure in the rebuttal.
>
> > You apply your test to every new datapoint. Pandeva et al only update after a small batch of new data is presented. Have you considered this? It might be computationally a bit more convenient, and I am curious how it would affect test power.
>
> In Appendix B, we discuss a minibatched version of our test where wealth updates are performed each time a batch of data is observed. It is a great question whether minibatches could improve power (i.e., be translated into the results about the growth rate of the underlying wealth process), and we plan to investigate this question in future work.

---

> > ### Comment · Reviewer_qWK6 · 2023-08-15
> > **Thanks -- no further questions**
> >
> > I thank the authors for their clarifications and adding additional experiments relative to Batch 2ST.  Also they have appropriately addressed my other questions which went beyond the initial submission.
> >
> > I conclusion I am raising my score from 7 to 8.

---

### Official Review · Reviewer_VGT6 · 2023-06-16

**Soundness:** 3 good
**Presentation:** 2 fair
**Contribution:** 3 good
**Rating:** 6
**Confidence:** 2

**Summary:**

The authors propose methods of two-sample and independence tests in the setting of sequentially released data. Theoretical and empirical evaluations of the proposed methods are included.

**Strengths:**

The authors propose algorithms for sequential two-sample and independence tests. The setting is interesting and important for practical applications. They provide theoretical results regarding the stop of the algorithm and the growth rate of the wealth. They also study the proposed methods numerically with synthetic and real-world data.

**Weaknesses:**

The readability can be improved. The introduction part is so long that I understand the contributions of the paper. I think the authors should split Section 1 into 2 sections: one is for explaining the motivation and the contribution, the other is for the problem setting.

**Questions:**

- In the paper, the initial values of $\lambda_t^{ONS}$ and $g_t$ are set as 0. Are there any strategies to set better initial values, or are there any empirical observations about that?

- What is $\sigma(Z_1,\ldots,Z_{t-1})$ in line 75? Does it mean the $\sigma$-algebra?

- In line 121, can we guarantee the existence of the minimizer $g_{\star}$? or is it just an assumption?

**Limitations:**

In the proposed algorithm, the computation is the same for all $t$. Thus, the estimated $\lambda_t^{ONS}$ and $g_t$ for small $t$ seem to perform worse than those for large $t$. It cannot be helped to a certain extent, but in the sequential setting, I think investigating some strategies to improve the performance for small $t$ would be important.

---

> ### Author Rebuttal · Authors · 2023-08-08
>
> We thank the reviewer for constructive feedback on the paper. Below, we address in detail issues that have been raised in the review.
>
> > Readability can be improved. The introduction part is so long that I understand the contributions of the paper. I think the authors should split Section 1 into 2 sections (motivation + contribution and problem setting).
>
> We thank the reviewer for suggestions about improving the readability of the paper. We have added more structure in the revised paper, specifically, we have added more subsections to Sections 1 and 2.
>
> > Initial values of $\lambda_t$ and $g_t$ are set as 0. Are there any better strategies / any empirical observations about that?
>
> Initializing $\lambda_t$ with 0 is not major since the online Newton step is used to update the betting fractions, and we simply used the default value specified in the earlier works. Initializing the predictor with a constant predictor is a bit more subtle: in our experiments, we trained all the models from scratch, but one can definitely use pre-trained models (e.g., weights of an image classifier trained on some other data) as initialization. We have added a remark about that point to the paper.
>
> > What is $\sigma (Z_1,\dots, Z_{t-1})$ in line 75? Does it mean the $\sigma$-algebra?
>
> This is correct. For $t\geq 1$, $\mathcal{F}\_{t}=\sigma (Z_1,\dots, Z_{t})$ stands for the sigma-field generated by $Z_1,\dots,Z_t$, and $\mathcal{F}_0$ stands for the trivial sigma-field. We have now clarified this part in the paper.
>
> > In line 121, can we guarantee the existence of the minimizer $g_\star$? or is it just an assumption?
>
> This part of the paper aims to build intuition. To define the oracle test, we only care about the risk of $g_\star$ (the optimal classifier in our class), and if the risk minimizer is not unique, $g_\star$ is chosen as any of the predictors that minimize the risk. When a classifier is trained online, then we do not need to assume convergence to $g_\star$, but the power will depend on the risk of the limiting classifier (whatever it is).
>
> > In the proposed algorithm, the computation is the same for all $t$. The estimated $\lambda_t$ and $g_t$ for small $t$ seem to perform worse than those for large $t$. It cannot be helped to a certain extent, but in the sequential setting, I think investigating some strategies to improve the performance for small $t$ would be important.
>
> Good point! The rule for updating $\lambda_t$ is indeed the same for all $t$. When the null hypothesis is false, the performance of a classifier $g_t$ clearly improves as more data are observed, and inferior performance in the beginning (when only a few points have been observed) is indeed expected. In practice, using pre-trained models (e.g., for image data) as initialization may definitely improve the performance during the early stages of testing. Another idea could be to bet slightly more conservatively for the first 20-30 (say) rounds, which is a heuristic that will affect the final regret guarantee or achieved growth rate.

---

> > ### Comment · Reviewer_VGT6 · 2023-08-16
> >
> > I have read the response. Thank you for your response.

---

### Official Review · Reviewer_FPjp · 2023-07-05

**Soundness:** 3 good
**Presentation:** 3 good
**Contribution:** 3 good
**Rating:** 6
**Confidence:** 3

**Summary:**

This paper proposes two sequential predictive two-sample tests based on betting, one is constructed by the payoff function $W\cdot \mathrm{sign}[g(Z)]$ for the misclassification risk, and the other is by the payoff function  $W\cdot g(Z)$ for the squared risk.  The limiting growth rate and the expected growth rate of both tests for an optimal $g^*\in G$ are given, and additionally, the same quantities of the squared-risk based test for an arbitrary $g\in G$ are given.

**Strengths:**

Originality\
The paper proposes and thoroughly analyzes the sequential predictive two-sample tests; to the best of my knowledge, the proposed tests and the theoretical results are new.

Quality\
The theoretical result is decent, which gives a growth rate comparison for the test based on the misclassification and the test based on the squared risk.

Clarity\
The paper overall presents well.

Significance\
The paper fills the blank that there is no formal construction of a classifier two-sample test using the betting framework.

**Weaknesses:**

1. The variance of the growth rate is not given and it is actually also important in the derivation of the testing power.

2. The expected growth rate is a result of optimal $\lambda$ instead of an arbitrary one.

3. The comparison with the likelihood ratio test (Alix Lhéritier, 2018) is missing in the real-data experiment.

**Questions:**

1. Why $\lambda$ is restricted to $[0, 1]$ instead of $[-1, 1]$. I can imagine a case where the payoff is negative, then using negative $\lambda$ would help increase the wealth in betting.

2. Is fitting the betting framework into the coin flip case the reason why the payoff functions are restricted to $[-1,1]$?

3. Why the ONS in algorithm 1 is different from the one in  (Cutkosky and Orabona, 2018) work and  (Shubhanshu Shekhar and Aaditya Ramdas, 2021), e.g., the update of $z_t$ and $\lambda_{t+1}^{ONS}$.

**Limitations:**

Please see the weakness section.

---

> ### Author Rebuttal · Authors · 2023-08-08
>
> We thank the reviewer for constructive feedback on the paper. Below, we address in detail issues that have been raised in the review.
>
> > Variance of the growth rate is not given ... it is actually important in the derivation of the testing power.
>
> We kindly disagree with the reviewer regarding this claim. To establish consistency of our test, it suffices to show that the growth rate of the wealth process (expected log wealth) is positive which we do: we show that the growth rate is lower bounded by a term proportional to the deviation of the risk of the limiting predictor from that of a naive predictor. Hence, as long as it is possible to learn a predictor with non-trivial prediction risk, our test is guaranteed to be consistent.
>
> > Expected growth rate is a result of optimal $\lambda$ instead of an arbitrary one.
>
> We kindly disagree with the reviewer regarding this claim. The lower bounds on the growth rate in Theorem 1 correspond to the case when the betting fractions are selected via the ONS strategy. The upper bounds indeed correspond to the optimal/oracle betting fraction $\lambda_\star$, and those are meant to quantify the best possible growth rate, and thus also show that the growth rate of ONS matches that of the oracle test. In summary, an arbitrary $\lambda$ would not work with our test: instead, we smartly set $\lambda_t$ online using ONS and the growth rate that this strategy achieves is at least (i.e. lower bounded by) that of the oracle $\lambda^*$ (which is unknown to us).
>
> > Comparison with the LR test (Alix Lhéritier, 2018) is missing in the real-data experiment.
>
> As we show through experiments, our predictive test (with CNNs) outperforms the one developed in [1] on image data. We omitted the comparison to the test of Lhéritier and Cazals [2019] since it has already been shown to be much inferior to the test developed in [1], specifically on multivariate data with localized differences between two distributions of interest (similar to the differences in our real data experiments). We have clarified that point in the revision.
>
> [1] ``Nonparametric Two-Sample Testing by Betting'', Shekhar and Ramdas, IEEE Transactions on Information Theory (to appear), 2023.
>
> > Why $\lambda$ is restricted to $[0,1]$ instead of $[-1,1]$. I can imagine a case where the payoff is negative, then using negative $\lambda$ would help increase the wealth in betting.
>
> Thanks for the great question! Technically, using the range $[-1,1]$ instead of $[0,1]$ is allowed, and none of the theoretical statements in our paper get affected. When ONS is used for selecting betting fractions, this translates into truncating $\lambda_t$ at -1/2 instead of 0. We tried both in our early experiments and did not observe any major differences in power, but truncating at 0 resulted in a bit more stable wealth processes, and hence, we used this option. If the alternative is true and a good classifier is learned online, then both of our payoffs will be positive on average, in which case it makes sense for $\lambda_t$ to be positive. We have added a short remark about this.
>
> > Is fitting the betting framework into the coin flip case the reason why the payoff functions are restricted to $[-1,1]$?
>
> Nonnegative (super)martingales starting at 1 are central in the testing by betting framework (in particular, it allows to instantiate Ville's inequality to guarantee ``time-uniform'' type-1 error control under the null hypothesis). At round $t$, the wealth is updated as $\mathcal{K}\_t = \mathcal{K}\_{t-1} \cdot (1+\lambda_t f_t(Z_t,W_t))$, and hence, restricting the payoff functions and betting fractions to $[-1,1]$ suffices to guarantee that the resulting process $(\mathcal{K}\_t)_{t\geq 0}$ is nonnegative. If the payoff is bounded in any other range, it can be transformed into $[-1,1]$, and we do not know how to design unbounded (on one side) bets that still result in nonnegative (super)martingales for our problem.
>
> > Why the ONS in algorithm 1 is different from the one in Cutkosky and Orabona ... e.g., the update of $z_t$ and $\lambda_{t+1}$.
>
> We thank the reviewer for reflecting upon that part. In the definition of $z_t$, there was indeed a typo: $z_t:=f_t/(1-\lambda_t f_t)$, which we have now fixed. Regarding the update rule for $\lambda_{t+1}$, there is a small change: instead of truncating at -1/2, we truncate at 0 (which is allowed and does not affect any theoretical claims of the paper). The reason behind this is the following: to maximize the power of the test, we aim to maximize the growth rate of the underlying wealth process, which corresponds to choosing $\lambda\_\star = \text{argmax}\_{\lambda\in [-1,1]}\mathbb{E}\log (1+\lambda f(Z_t,W_t))$. For a given predictor $g$, consider the payoff corresponding to the squared risk: $f(Z_t, W_t) = W_t\cdot g(Z_t)$, where $W_t\in\\{-1,+1\\}$ and $g_t\in [-1,1]$. Assuming the 2ST null is false and $g$ is a predictor with non-trivial prediction risk, we have that: $\mathbb{E}f(Z_t,W_t)>0$, in which case it is easy to see that $\lambda_\star$ has to be positive. In early simulations, we also did not observe any major differences between the two truncating options. We have clarified that part in the revision.

---

> > ### Comment · Reviewer_FPjp · 2023-08-11
> >
> > Thank you for your reply. \
> > **Regarding the variance of growth rate**\
> > I understand the test's consistency is irrelevant to the variance of the growth rate. However, I was referring to the testing power, which is a finite-sample result. That is related to the variance of the growth rate.
> >
> > As for the comparison with [1], did you use CNN for your method and use a kernel to construct the sequential MMD test for [1]?

---

> > > ### Comment · Reviewer_FPjp · 2023-08-11
> > >
> > > I re-read the paper, especially the experiments and conclusion. If I understand that correctly, the gain of testing power is attributed to the use of a predictive model over the kernel instead of the form of constructed statistic (e.g., with the payoff $W g(Z)$)? In fact, I found the gap between the proposed work and [1] is smaller than I thought. The sequential MMD test, which is used as an example for the test by betting in [1], can also be simply constructed with a predictive model by replacing $g$ with a predictive model; although $g$ might not by contained in the RKHS anymore. Hence, the superiority of the proposed method over the ''kernel'' test counterparts seems to be questionable to me.

---

> > > ### Author Response · Authors · 2023-08-12
> > >
> > > > Regarding the variance of growth rate: I understand the test's consistency is irrelevant to the variance of the growth rate. However, I was referring to the testing power, which is a finite-sample result. That is related to the variance of the growth rate.
> > >
> > > Variance of the log wealth is not a metric that has ever been considered before in any paper on betting from the original (non-testing) works in information theory like Kelly (1954), Breiman (1963), Cover (1970s), or in any paper on using betting for statistical hypothesis testing (i.e., papers cited in our work). It is the variance of the payoffs that is important, not the variance of the growth rate (i.e., not the variance of the log payoffs): in particular, this is the reason why we have two terms in our bounds on the growth rate and why under the ''low-variance'' regime, we get a faster growth rate (the expected margin is not squared). We have finite sample guarantees on the rate of growth of wealth (due to the use of nonasymptotic regret bounds in our proofs), but we have stated the asymptotic growth rate because it is more interpretable.
> > >
> > > Regarding additional finite-sample results, we now also have bounds for the stopping time $\tau$, namely $P(\tau>t)$ and $\mathbb{E}[\tau]$, where an upper bound on the latter easily follows from the former bound (since $\mathbb{E}[\tau] = \sum_{t=0}^\infty P(\tau>t)$) for the case when the 2ST null is false. For simplicity, assume that the same classifier $g$ (e.g., the Bayes-optimal classifier) is utilized for betting using the payoff: $f(Z_t,W_t)=W\cdot g(Z_t)$. We show that:
> > >
> > > $\mathbb{E}[\tau] \leq O\Big( \Big( \frac{1}{\mathbb{E}[W \cdot g(Z)]} + \frac{1}{\sup_{s\in[0,1]}(1-R_s(sg))} \Big)\cdot \log(1/\alpha) \Big)$,
> > >
> > > meaning that the expected stopping time is upper bounded by the sum of the reciprocals of the expected margin of a classifier and the deviation of its (optimized) squared risk from the worst-case value. We get the squared risk term exactly because we account for the variance of the payoffs (otherwise, the upper bound is proportional only to the reciprocal of the *squared* expected margin of a classifier, which is always worse). We have added this result with the proof to the Appendix, and we are more than happy to provide a short proof sketch!
> > >
> > > > As for the comparison with [1], did you use CNN for your method and use a kernel to construct the sequential MMD test for [1]?
> > >
> > > This is correct.
> > >
> > > > I re-read the paper, especially the experiments and conclusion. If I understand that correctly, the gain of testing power is attributed to the use of a predictive model over the kernel instead of the form of constructed statistic (e.g., with the payoff $W_tg(Z_t)$)? In fact, I found the gap between the proposed work and [1] smaller than I thought. The sequential MMD test, which is used as an example for the test by betting in [1], can also be simply constructed with a predictive model by replacing it with a predictive model; although might not be contained in the RKHS anymore. Hence, the superiority of the proposed method over the ''kernel'' test counterparts seems to be questionable to me.
> > >
> > > While the two betting-based tests are closely related (which we highlighted at the end of the Introduction), our paper actually constructs a different betting game from that in [1]. Both papers are indeed similar from the standpoint of using a chosen distance measure between distributions (e.g., kernel-MMD in [1], or the squared risk of a classifier in our work) as a starting point for designing effective betting strategies. However, our paper is not just about constructing a valid test, but characterizing its quality: we explicitly relate the growth rate and consistency to interpretable and intuitive metrics that are associated with classifiers, namely their risks and margin.
> > >
> > > When we use the squared risk as a measure of distance, the resulting bets indeed happen to take a similar form as in the sequential kernel-MMD test from [1]. However, our theoretical results provide practical suggestions for designing powerful predictive 2STs, e.g., we show that training a classifier via direct margin maximization / minimizing the hinge loss (which may be hinted at by the distance measure in [1]) will hurt the power of the resulting 2ST, i.e., the growth rate of the wealth. Further, we also relate our test to earlier sequential predictive 2STs and show how we address their limitations.
> > >
> > > Last, we never argued that either of the approaches: kernel- or classifier-based, is superior to the other and confirmed that across different experimental setups. Our motivation was not to develop a predictive test that should replace a kernel one but rather to complement the existing kernel approach with a new method that may be better suited for high-dimensional or structured data (like images).

---

> > > > ### Comment · Reviewer_FPjp · 2023-08-16
> > > >
> > > > Thank you for your clarification. I acknowledge the main contribution of this work is a formal formalization of predictive two-sample testing by betting and deriving sound theorems about the test. A follow-up question about the ONS: the ONS proposed in the original paper is to minimize a reward. It is plausible to use their ONS algorithm here to maximize the reward (the wealth). However, as far as I understand, the ONS in your algorithm swaps the max and min operations in the original ONS; I am not sure if this could still maintain the original asymptotic optimality of the interested parameter. A safe way to maintain the same result of the original ONS from my perspective, which seems to be different from the ONS you implement, is to plug $-\lambda_t$ into the original ONS, update $-\lambda_t$ to $-\lambda_{t+1}$ and then use the negation of $-\lambda_{t+1}$. Can you comment on this confusion about ONS?

---

> > > > > ### Author Response · Authors · 2023-08-16
> > > > >
> > > > > Sure! We thank the reviewer for pointing out another typo in the description of the ONS algorithm. Before diving into the details, we would like to note a correct implementation of the algorithm is used in our experiments, and hence, all the results still stand. [1] applies ONS for coin betting and defines the wealth updates as $\mathrm{Wealth}\_t=\text{Wealth}_{t-1} \cdot(1-g_t v_t)$. Here, $g_t$ denotes the observed outcomes (which is $-f_t$ in our notation), and $v_t$ is our $\lambda_t$. A correct version of the ONS for updating betting fractions according to our notation (which matches that in [1]) is stated below:
> > > > >
> > > > > 1. Observe $f_t \in [-1, 1]$;
> > > > > 2. Set $z_t := \frac{-f_t}{1+\lambda_{t}^{ONS} f_t}$;
> > > > > 3. Set $ a_t := a_{t-1} + z^2_t$ ;
> > > > > 4. Set $\lambda_{t+1}^{ONS} := \min(1/2, \max(0, \lambda_{t+1}^{ONS} - \frac{2}{2−\log 3} \cdot \frac{z_t}{a_t}))$;
> > > > >
> > > > > We have updated the paper to include the correct version of the algorithm. You can also verify that our experiments indeed use the correct algorithm by checking our code: in the supplement, if you open /code/utils/testing.py and search for “ONS” as a betting scheme (self.bet_scheme), you will find the following in lines 135--139:
> > > > >
> > > > > ` grad = self.run_mean/(1+self.run_mean*self.opt_lmbd) `
> > > > >
> > > > > ` self.grad_sq_sum += grad**2 `
> > > > >
> > > > > ` self.opt_lmbd = max(0, min(self.truncation_level, self.opt_lmbd + 2/(2-np.log(3))*grad/self.grad_sq_sum)) `
> > > > >
> > > > > The above snippet matches the pseudo-code above and uses the correct signs (we moved the negation from the numerator in step 2 to the update rule in step 4 which corresponds to solving the maximization problem, i.e., grad is the derivative of $h(\lambda) = \log(1+\lambda\cdot f_t)$ and a plus has is a sign in the last line).
> > > > >
> > > > > [1] “Black-Box Reductions for Parameter-free Online Learning in Banach Spaces”, Cutkosky and Orabona, 2018.

---

> > > > > > ### Comment · Reviewer_FPjp · 2023-08-16
> > > > > >
> > > > > > Thank you for your clarification. My concerns are largely addressed. I am happy to increase the score from 5 to 6.

---

### Official Review · Reviewer_ssj4 · 2023-07-07

**Soundness:** 3 good
**Presentation:** 1 poor
**Contribution:** 2 fair
**Rating:** 4
**Confidence:** 4

**Summary:**

The work considered the problems of sequential nonparametric two-sample and independence
testing. The researchers propose a novel approach, which overcomes the issues of kernel-based testing, such as finding an appropriate kernel for high-dimensional or structured data like text and images. The authors empirically demonstrate the superiority of their tests over
kernel-based methods in structured settings. Furthermore, the tests remain valid and powerful even when the data distribution drifts over time.

**Strengths:**

1. The authors propose a prediction-based betting strategies to help alleviate problems associated with kernel selection and adaptability with high-dimensional or structured data.

2. The authors provide compelling empirical evidence that their method outperforms existing kernel-based tests, especially in the context of structured settings.

3. The author provides the theoretical analysis about the properties of the proposed method.

**Weaknesses:**

1. The authors should provide more technical details about the principle of testing by betting strategies. This is the key concepts in the paper but the author does not elaborate it too much in the paper.

2. The paper provides a high-level overview of the research making it challenging for general readers to comprehend.

3. The presentation is poor is a way mixed with methodology, theory, and numerical study in a single section (e.g. Section 2).

4. It is uncommon that one paper focused on two testings: the two-sample test and the independence test.


**Questions:**

1. Does the proposed method have any potential limitations on data-dependency due to the use of nonparametric techniques?

2. Could the authors clarify how your method can retain its power when data distribution drifts over time?

3. It is not clear how the proposed techniques can be scalable to larger datasets?

**Limitations:**

The authors do not mention the limitations of this work.

---

> ### Author Rebuttal · Authors · 2023-08-08
>
> We thank the reviewer for constructive feedback on the paper. Below, we address in detail issues that have been raised in the review.
>
> > ...more technical details about the principle of testing by betting strategies...key concept in the paper but the author does not elaborate it too much in the paper...
>
> We have added more technical details (e.g., clarifying that under the null hypothesis, the resulting wealth process is a nonnegative martingale starting at 1, which is the key to invoking Ville's inequality to justify the type-1 error control) as well as a more intuitive explanation regarding the principle of testing by betting to the revision.
>
> > A high-level overview of the research makes it challenging for general readers to comprehend the paper.
>
> To improve the comprehension for general readers, we made the following updates in the revision: (a) a more detailed explanation of the principle of testing by betting (improving on both technical and intuitive sides), (b) a more comprehensive review of the most closely related works (making it easier to disseminate the contributions of our work).
>
> > The presentation is poor...mixed methodology, theory, and numerical study in a single section (e.g. Section 2).
>
> To improve the presentation of the paper, we have separated experiments from the methods and theory in the revision.
>
> > Uncommon that one paper focuses on two-sample and independence testing
>
> We kindly disagree with the reviewer that this point represents a weakness of our paper. While it is less common for a single paper to consider both problems, the two are closely related: an algorithm for one can be used for the other and vice versa (i.e., each problem reduces to the other, but this reduction may not be the optimal way to solve the problems). In our paper, we use the connection between the two problems explicitly: we design and evaluate sequential predictive tests for both settings.
>
> > Any potential limitations of the method on data dependency due to the use of nonparametric techniques?
>
> We do not see any limitations due to the nonparametric aspect currently, but perhaps we have not fully understood what the reviewer may be hinting at.
>
> > Clarify how the method can retain its power when data distribution drifts over time?
>
> Under the alternative, the power of our test depends on the performance of a classifier according to misclassification/squared risk. If the distribution drifts over time, then models updated via online gradient descent may retain their predictive power. In such cases, our test will still have high power despite the present distribution drift. Of course, if the distribution shifts adversarially, the method will not have power, but neither will any other method. So the implicit goal is to maintain type-1 error control under the null despite shifting distributions (beyond the iid assumption typically made in the literature) while retaining power against reasonable (but not all) distribution drifts.
>
> To put this into context, we conducted the following experiment (the results have also been added to the revision) where we consider four settings:
>
> (a) $P=Q= \mathcal{N}(0,1)$, i.e., the 2ST null is true,
>
> (b) $P= \mathcal{N}(0,1)$ and $Q= \mathcal{N}(0.5,1)$, i.e., the 2ST null is false,
>
> (c) up to time $t=250$, $P^{(t)}=Q^{(t)}= \mathcal{N}(0,1)$, but for $t > 251$, we have $P^{(t)}= \mathcal{N}(0,1)$ and $Q^{(t)}= \mathcal{N}(0.5,1)$, i.e., there is a distribution shift and the 2ST null is false,
>
> (d) up to time $t=250$, $P^{(t)}=Q^{(t)}= \mathcal{N}(0,1)$,  but for $t > 251$, $P^{(t)}=Q^{(t)}= \mathcal{N}(0.5,1)$, i.e., there is a change in distribution yet the 2ST null is still true.
>
> In all settings, we monitor the tests until $T=2000$ observations. We use a standard logistic regression model as an underlying predictor with weights updated via online gradient descent. We have uploaded the figure with the respective rejection rates to the allowed PDF file. In a nutshell, our test controls the type-1 error whenever the 2ST null is true even if there is a shift in distribution, and retains high power if the alternative is true.
>
> > It is not clear how the proposed techniques can be scalable to larger datasets.
>
> The main computational burden of our test lies on a selected classifier and learning algorithm. The data are processed one at a time in a sequential fashion, so it is only the per-point update cost that is critical. Hence, for many models trained online, using versions of stochastic gradient descent, like neural nets, our test will scale perfectly well to larger datasets. In fact, one of the advantages of our test over the batch tests (which would require thinking about the sample size in advance) is that our test adapts to the complexity of a problem at hand. Hence, if the null is false, our test may stop after processing a few hundred observations, whereas its batch counterpart will require training using an unnecessarily large sample.

---

> > ### Comment · Reviewer_ssj4 · 2023-08-16
> >
> > I have read the response. Thank you for your response.

---

### Official Review · Reviewer_ryYs · 2023-07-08

**Soundness:** 4 excellent
**Presentation:** 4 excellent
**Contribution:** 4 excellent
**Rating:** 7
**Confidence:** 4

**Summary:**

Kernel-based nonparametric two sample and independence tests performance can break down in the cases of complex data like images. The paper proposes sequential two sample and independence tests based on the misclassification rate.

Compared to kernel-based tests, the proposed tests, which use CNNs, reject the null hypothesis faster than sequential kernel-based two sample and independence tests on image data while still controlling the type I error rate. The power comparison between the two approaches is somewhat inconclusive.

**Strengths:**

The proposed approach is quite general and well suited to complex data like images that can be modeled with DNNs.

The work is well supported theoretically.

The empirical results argue favorably for the approach.

The paper is well written.

**Weaknesses:**

Some discussion of the practical computational complexity of the approach compared to kernel based tests would improve the paper.

An empirical comparison to kernel-based tests on simpler data would highlight the advantages/disadvantages of each approach.


**Questions:**

Optimizing the bandwidths is one drawback of kernel tests - how much hyperparameter tuning was required to achieve the performance reported?

Have the authors compared the approach to kernel-based tests on simpler data?

How do the runtimes compare to kernel-based tests?

**Limitations:**

Could be expanded

---

> ### Author Rebuttal · Authors · 2023-08-08
>
> We thank the reviewer for constructive feedback on the paper. Below, we address in detail issues that have been raised in the review.
>
> > Some discussion of the practical computational complexity of the approach compared to kernel-based tests would improve the paper. How do the runtimes compare?
>
> Below, we assume that the tests are deployed for $d$-dimensional data. At time $T$, the total accumulated computation for the kernelized test is $O(dT^2)$. For our test, the answer depends on the chosen classifier and learning algorithm. For example, using logistic regression in combination with gradient descent for updating model parameters results in cheap updates and payoff evaluation (both are $O(d)$ at each round, and hence the total accumulated computation at time $T$ is $O(dT)$). For $k$-NN classifier, no parameters have to be updated, yet evaluating payoffs becomes more expensive with a growing sample size, resulting in the total accumulated computation of $O(k dT^2)$ at time $T$. For more complex models like neural nets, runtime depends on the chosen architecture: the total accumulated computation at time $T$ is $O((cB+F)T)$, where $F$ and $B$ are the costs of forward-propagation and back-propagation steps respectively and $c$ is the number of back-propagation steps applied after processing the next point (the exact cost depends on the architecture). We have added a paragraph about the runtime comparison to the paper.
>
> > Empirical comparison to kernel tests on simpler data would highlight the advantages/disadvantages of each approach.
>
> Overall, our empirical findings illustrate that neither of the approaches (kernel-based vs predictive) dominates the other and that our new approaches are surprisingly versatile across settings. To elaborate, some additional simulations that highlight the advantages/disadvantages of kernel approaches versus our new methods are already in Appendix E. While it may seem a natural guess that kernel methods would perform better for ``simpler'' unstructured data, our results in Appendix E suggest that this is not always the case. If there is any particular data distribution you would like us to add, we would be happy to do so.
>
>
> > Optimizing the bandwidths is one drawback of kernel tests - how much hyperparameter tuning was required to achieve the performance reported?
>
> For simulations on synthetic data, we utilized the knowledge of the underlying distributions to estimate kernel hyperparameters, taking those to be inversely proportional to the second moment of the random variables (details are provided in Appendix E). For simulations on real data, we utilized the median heuristic (standard practice) to estimate these hyperparameters. For our method, we applied minimal hyperparameter tuning: (a) for MLPs and CNNs, we committed to a single architecture throughout the experiments and used early stopping for regularization, (b) for experiments with $k$-NN classifier, there are no hyperparameters to be tuned except for the number of neighbors, and we used the square root rule (standard practice). In fact, one important advantage of our sequential test is that various design choices (increasing/decreasing regularization, changing neural network architecture) can also be updated on-the-fly.

---

### Author Rebuttal · Authors · 2023-08-09

Dear Reviewers,

We wanted to thank you for your time and for your valuable feedback! We hope that our responses address many/most of the existing concerns. We also attach a PDF file that contains the results of several additional experiments you asked for. If you have any additional questions, we would love to hear those from you and engage in a discussion. Looking forward.

Sincerely,
The authors.

---

### Decision · Program_Chairs · 2023-09-21

**Decision:**

Accept (poster)

**Comment:**

The focus of the submission is sequential two-sample and independence testing. Using the principle of testing by betting, the authors present a novel sequentially learnt predictor-based hypothesis testing framework (Algorithm 2) which is shown to be level-alpha and consistent (Theorem 2), alongside with the characterization of the growth of the underlying oracle wealth process (Theorem 1). The practicality of the method is showcased on the KDEF and MNIST datasets.

Sequential hypothesis testing is a key problem in data science. Constructing new, flexible, principled, and practically-efficient techniques in the domain, complementing existing kernel-based approaches is without doubt relevant to the community. The submission nicely achieves this goal, as it was detailed by the reviewers.